

# Consistency between the Strain Rate Model and ESHM20 Earthquake Rate Forecast in Europe: insights for seismic hazard

Donniol Jouve Bénédicte[1], Socquet Anne[1], Beauval Céline[1], Piña Valdès Jesus[2], and Danciu Laurentiu[3]

[1]Univ. Grenoble Alpes, Univ. Savoie Mont Blanc, CNRS, IRD, Univ. Gustave Eiffel, ISTerre, 38000 Grenoble, France
[2]Departamento de Ciencias Geodésicas y Geomática, Escuela de Ciencias y Tecnología, Universidad de Concepción Campus, Los Ángeles, Chile
[3]Swiss Seismological Service, ETH Zürich, Zürich, Switzerland

**Correspondence:** Donniol Jouve Bénédicte (benedicte.donniol@ikmail.com)

**Abstract.** The primary aim of this research is to investigate how geodetic monitoring can offer valuable constraints to enhance the accuracy of the source model in probabilistic seismic hazard assessment. We leverage the release of geodetic strain rate maps for Europe, as derived by Piña-Valdés et al. (2022), and the ESHM20 source model by Danciu et al. (2024) to compare geodetic and seismic moment rates across Europe, a geographically extensive region characterized by heterogeneous seismic activity. Seismic moment computation relies on the magnitude-frequency distribution proposed in the ESHM20 source model logic tree, which is based on earthquake catalogs and fault datasets. This approach allows us to account for epistemic uncertainties proposed in ESHM20. On the geodesy side, we meticulously calculate the geodetic moment for each zone, considering associated epistemic uncertainties. Comparing the distributions of geodetic and seismic moments rates at different scales allows us to assess compatibility. The geodetic moment rate linearly depends of the seismogenic thickness, that is therefore a pivotal parameter contributing to the uncertainty. In high-activity zones, such as the Apennines, Greece, the Balkans, and the Betics, primary compatibility between seismic and geodetic moment rates is evident. However, local disparities underscore the importance of source zone scale; broader zones enhance the overlap between geodetic and seismic moment rate distributions. Discrepancies emerge in low-to-moderate activity zones, particularly in areas affected by Scandinavian Glacial Isostatic Adjustment, where geodetic moment rates exceed seismic moment rates significantly. Nevertheless, in some zones where ESHM20 recurrence models are well-constrained, by either enough seismic events in the catalogue or mapped active faults, we observe an overlap in the distributions of seismic and geodetic moments, suggesting the potential for integrating geodetic data even in regions with low deformation.

## 1 Introduction

Nowadays, source models in up-to-date probabilistic seismic hazard studies are based both on past seismicity and active tectonics datasets. For example, the source model logic tree in the European Seismic Hazard Model 2013 (Woessner et al. (2015)) and its update the European Seismic Hazard Model 2020 (Danciu et al. (2024)) include two main branches, an area source model and a fault model. In regions where active faults are rather well-characterized, they must be accounted for in the hazard estimations (e.g. Stirling et al. (2012); Field et al. (2014); Beauval et al. (2018)). Fault models are mostly based



on geologic information, covering much larger time windows than the available earthquake catalogs. Fault models thus bring
insights on the generation of earthquakes that complement the catalog-based earthquake forecasts. However, fault databases are
known to be incomplete, even in the best characterized regions, and earthquakes may occur on unknown faults, as demonstrated
by several earthquakes in the past (e.g. the two 2002 Mw 5.7 Molise earthquakes (Valensise et al. (2004)) in Italy; or the Darfield
Mw 7.1 earthquake in New Zealand (Hornblow et al. (2014))).

The use of geodetic data in the development of source models has been limited up to now, although deformation rates
based on velocities from the Global Navigation Satellite Systems (GNSS) constitutes a promising perspective for constraining
earthquake recurrence models (Jenny et al. (2004); Shen et al. (2007)). GNSS stations measure the present-day displacements
at the surface of the earth. A convenient way to characterize the ground deformation is to invert the surface velocities measured
by GNSS to compute strain rate maps, that are independent from the reference frame. The accuracy of the estimated strain
rates depends on the spatial density of GNSS stations, on the quality of the sites, and on the duration of the records (Mathey
et al. (2018)). Along major interplate faults, such as subduction zones or lithospheric strike slip faults, interseismic velocities
measured by GNSS are now commonly used to constrain the slip deficit on the fault associated with locking in between large
seismic events (also referred to as interseismic coupling). In such highly active tectonic boundary regions, the interseismic
slip deficit may be combined with the earthquake catalog to constrain earthquake recurrence (Avouac (2015); Mariniere et al.
(2021)). In plate interiors, where the faults are moving at low slip rates and where the fault mapping might be not enough
developed, strain rate models can provide constraints on the seismic potential.

Indeed, the tectonic loading recorded by geodesy should be proportional to the energy released during earthquakes, under the
assumption that the earth's crust behaves elastically. If this assumption is true and if other factors such as aseismic deformation
are not significant, then the rate at which energy is released during earthquakes (represented by the seismic moment rate) and
the rate at which tectonic forces build up between earthquakes (represented by the geodetic moment rates) should be equal
(Stevens and Avouac (2021)). This balance can be used to constrain magnitude-frequency distributions. In the last 30 years,
a number of studies have analyzed the catalog-based magnitude-frequency distributions with respect to the tectonic loading
measured by geodesy. In the Hellenic arc, Jenny et al. (2004), found that the maximum magnitudes required for the earthquake
recurrence models to be moment-balanced were unrealistic and concluded that a large part of the strain is released in aseismic
processes. In the India-Asia collision zone, Stevens and Avouac (2021) highlighted a correlation between earthquake rates
and strain rates. They established moment-balanced recurrence models that fit both past seismicity and the geodetic moment,
bounded by maximum magnitudes compatible with those expected in the region.

Determining the extent to which the methods used to study highly active tectonic regions can be applied to areas with lower
levels of seismic activity is an open research question. The present study is at the scale of the whole European continent, that
is very heterogeneous in terms of tectonic activity. Southern Europe, with regions such as the Apennines, Greece and Turkey,
is characterized by a high seismic activity and significant tectonic deformation (Nocquet (2012); de Vicente and Vegas (2009);
whereas northern and central Europe is characterized by a low to moderate seismic activity (Kierulf et al. (2021); Lukk et al.
(2019)). We take advantage of two new studies performed at the scale of Europe: the release of the new probabilistic seismic
hazard model for Europe (ESHM20, Danciu et al. (2021)); and the strain rates models computed by Piña-Valdés et al. (2022)



as presented in Figure 1. Our objective is to compare the ESHM20 earthquake rate forecast with the deformation rates obtained

from the GNSS velocities, giving special attention to the estimation of uncertainties.

In a first step, we present the datasets and methods used to compute the seismic and geodetic moments integrated in space and time and to explore the uncertainties. Next, we compare the estimated seismic and geodetic moments in the different seismogenic source zones of ESHM20 that covers the Euro-Mediteranean region. We then discuss the parameters that influence the most the compatibility in both high and low-to moderate seismic activity.

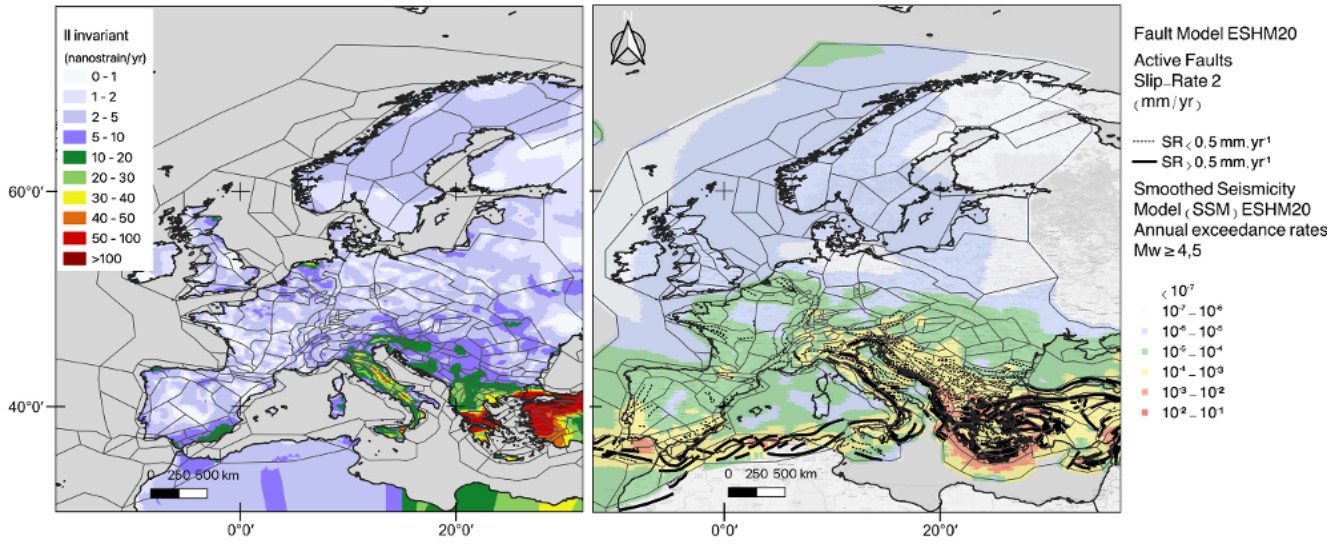

**Figure 1.** Strain rate model for Europe and ESHM20 earthquake rate forecast (smoothed seismicity and fault model branch). (a) II invariant of the strain rate tensor (Piña-Valdés et al. (2022)), with area sources of ESHM20 superimposed; (b) Smoothed seismicity model, earthquakes rates $M_W \geq 4.5$, faults included in the model are superimposed (Danciu et al. (2021))




## 1.1 Seismic moment: moment distribution associated with the ESHM20 source model logic tree

ESHM20 aims at delivering seismic hazard levels throughout Europe, using harmonized datasets and applying homogeneous methodologies (Danciu et al. (2021)). The regional hazard model consists of two main components a seismogenic source model and a ground-motion characteristic model. The present study deals with the seismogenic source model and its components. The earthquake rate forecast includes all earthquake types, i.e., crustal, deep, and subduction earthquakes. In this paper we focus on the contribution of crustal shallow seismogenic sources, that can be straightforward compared to surface strain rate.

The ESHM20's seismogenic source model is based on several updated datasets (Danciu et al. (2021)): an earthquake catalog, covering the time window 1000-2014, including both historical (Rovida et al. (2022)) and instrumental (Lammers et al. (2023)) periods, and a fault database including potentially active faults, with their geometry and geologic or geodetic slip rates (European Fault-Source Model 2020 EFSM20, Basili et al. (2023)). The seismogenic source model logic tree accounts for alternative seismogenic source models to capture the spatial and temporal variability of the earthquake rate forecast in Europe. It includes two main branches : an area source model and a hybrid model that combines active shallow faults with a background smoothed seismicity with an adaptive kernel (Danciu et al. (2021)).

The area source model consists of cross-border harmonized seismogenic sources which geometry is guided by seismotectonic evidence such as potentially active faults, geologic features, seismicity pattern (Danciu et al. (2021)). For every area source, a Gutenberg magnitude-frequency distribution (Gutenberg and Richter (1944)) has been established from the earthquake catalog taking into account time windows of completeness. Two alternative models have been considered to account for the uncertainty in forecasting earthquake rates in the upper magnitude range:

- a magnitude-frequency distribution truncated at a maximum magnitude $M_{max}$, corresponding to form 2 in Anderson and Luco (1983) :

$$N(m) = 10^{a-b*m} + 10^{a-bM_{max}} \text{ for m} \leq M_{max} \tag{1}$$

- a tapered Pareto distribution (Kagan (2002)) which includes a bending of the recurrence model from a magnitude called the corner magnitude ($M_c$).

As an alternative to the area source model, the hybrid model consisting of active crustal faults combined with off-fault smoothed seismicity (Fig 2). For each fault, a moment-balanced magnitude-frequency distribution has been established, that accommodates the moment inferred from the slip rate and the geometry of the fault, assuming moment conservation principle. The maximum magnitude is obtained applying the Leonard (2015), scaling relationship to the length, the width and the area of the fault (Basili et al. (2023)). The smoothed seismicity model is based on the unified earthquake catalog and an adaptive kernel obtain for the entire region. To avoid double-counting, a buffer zone is applied around each fault (see Danciu et al. (2021)).

The source model logic tree explores the uncertainty on the definition of the maximum (or corner) magnitude both in the area source model and in the fault model (Figure 2). For the area source model, the uncertainty on the estimation of a- and b-values is also considered (Gutenberg-Richter model branch). For the fault model, the uncertainty on the slip rate estimates is explored. Overall, the exploration of the logic tree leads to 21 alternative recurrence models, with different weights.



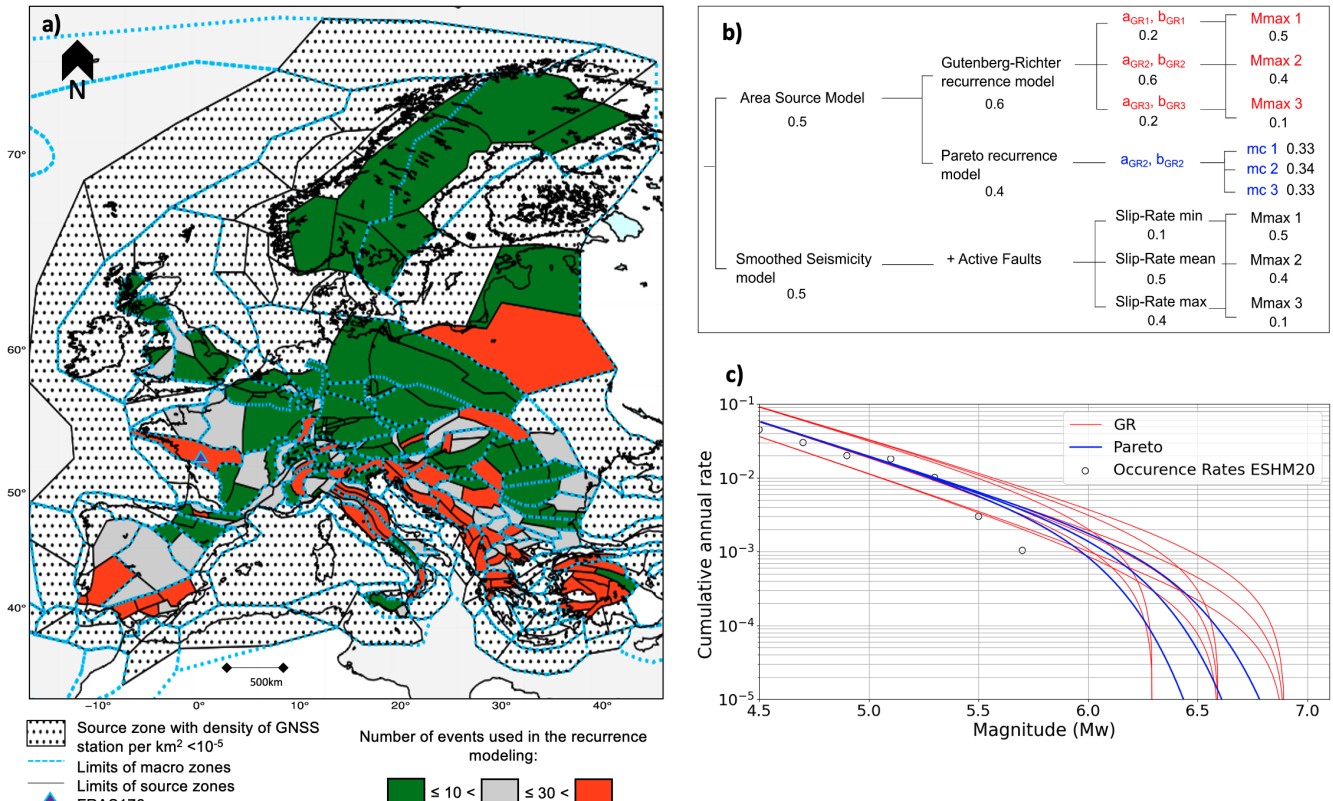

**Figure 2.** ESHM20 source model (Danciu et al. (2021)): (a) area sources (black polygons), and larger macrozones (dashed blue) used to infer the b-value in regions with poor earthquake data; orange: sources with at least 30 events used to establish the recurrence model, green: with less than 10 events, black dots: area sources not considered in the study (poorly constrained strain rates). (b) Source model logic tree, with the weights associated to the different branches. (c) Alternative earthquake recurrence models for the example source zone FRAS176 (southern Brittany in France, blue triangle), colors correspond to the branch combinations in the area source model, Fig. 2b

For every area source zone, 21 alternative estimates for the seismic moment are computed from the 21 alternative magnitude frequency distribution. Considering the recurrence models Gutenberg-Richter and Pareto, the total annual moment rate corre-
sponds to the integral under the curve in terms of moment. In the case of the Gutenberg-Richter model (form 2 in Anderson and Luco (1983)), the following equation is used (Mariniere et al. (2021)) :

$$\dot{M}_{0S} = \frac{b}{(c-b)} * 10^{a+d+(c-b)M_{max}} \tag{2}$$

in $N.m.yr^{-1}$, with $c = 1.5$ and $d = 9.1$ the parameters used in the calculation of the seismic moment from the moment magnitude, Hanks and Kanamori (1979).

To compute the annual seismic moment rate from the smoothed seismicity and fault model, we sum the seismic moments associated to every spatial cell within the area source zone (one magnitude-frequency distribution per cell). When a fault



straddles several zones, the seismic moment associated to the source zone is proportional to the length of the fault within the source zone.

For each source zone, a distribution of 21 seismic moments is obtained, representative of the uncertainties considered in the
ESHM20 seismogenic source logic tree. A weighted mean seismic moment is calculated considering the weights associated to every branch combination (Fig 2). Besides, approximate $16^{th}$ and $84^{th}$ percentiles are inferred from the discrete distributions.

## 1.2   Geodetic moment computation from strain rates maps and uncertainty exploration

Our aim is to use strain rates evaluated at the scale of Europe to estimate the geodetic moment rate within every area source of the ESHM20 source model. To achieve this goal, we start from the work done by Piña-Valdés et al. (2022). They combined
ten GNSS velocity fields with different spatial coverage in Europe. After filtering the velocity field obtained to remove stations with highest uncertainties, they applied the VISR algorithm (Shen et al. (2015)) to derive a strain rate map for Europe (a best estimate model). The algorithm VISR calculates horizontal strains through interpolation of a geodetic velocity field. It is an undetermined inverse problem; the algorithm uses as inputs the discretized geodetic observations and delivers smoothed distributed strain rates. Key decisions need to be taken on the exact weighting scheme to apply, that may impact the interpolation
and the final strain rate estimates. In our case, rather than a best estimate, we need a distribution for the geodetic moment rate that is representative of the uncertainties.

### 1.2.1   Uncertainties on the strain rate estimates

Ideally, only the stations with the best constrained velocity estimates should be included for deriving strain rates, however a compromise must be obtained between discarding poorly constrained stations and keeping a reasonable number of stations
for the analysis. Piña-Valdés et al. (2022) have classified the 4863 available stations into 4 categories A, B, C, and remaining stations, depending on their noise level that increase the uncertainty on the velocity (increasing the uncertainty from A class station ahead). To derive their best model, they finally decided to include all stations falling into categories A, B and C. Here, we are interested in quantifying how much this decision impacts the strain rate estimates and we explore the uncertainty related to the use of only class A stations (3377), of both A and B stations (4091), or all stations A, B and C (4468).

For the strain rates to be reliable, anomalous velocities must be identified and removed from the combined velocity field. Piña-Valdés et al. (2022) proposed to detect outliers based on an analysis of the spatial consistency of the velocities. For every station, the distribution of the velocities within a circular region around the station is obtained; stations with velocities in the tails of the distribution are considered outliers. Piña-Valdés et al. (2022) tested 4 different radius (50, 100, 150 and 200km) and showed that when the radius is increased, the number of outliers decreases. They used 150km for deriving their best estimate
model, considering this radius a compromise between the number of stations left (4238) and a reduction of the variance obtained on the final solution. Here, we keep track of the uncertainty associated with this decision, and we use alternatively the 4 different radii to evaluate strain rates.

While applying the algorithm VISR, a number of decisions are required that may impact horizontal strain rates estimates. Shen et al. (2015) show that the distance-dependent weighting can be achieved by employing either a Gaussian or a Quadratic





decay function, and that for the spatially-dependent weighting either an Azimuthal weighting or a Voronoi cell area weighting function can be applied. Another crucial parameter is the weighting threshold, which governs the smoothing of the inversion process. Here we include in the analysis both the uncertainty on the smoothing function and on the spatially dependent weighting, as well as three alternative weighting thresholds values (6, 12 and 24; see Shen et al. (2015)).

### 1.2.2    Estimation of the geodetic moment rate within an area source zone

For each area source of the EHSM20 model, we determine a distribution for the geodetic moment rate. Figure 3 illustrates the different steps for the source zones in Northwestern France.

First, for each component of the strain rate tensor ($\dot{\varepsilon}_{xx}$, $\dot{\varepsilon}_{yy}$, $\dot{\varepsilon}_{xy}$), we determine the mean component from all grid cells falling within the source zone (Figs. 3a and 3b) :

$$\overline{\dot{\varepsilon}_{xx}}^2 = \frac{\sum_{i=1}^{ncells} \dot{\varepsilon}_{xx}(i)}{n} \tag{3}$$

Then we calculate the principal components (eigenvalues) of the strain rate tensor within the area source :

$$\overline{\dot{\varepsilon}_{max}} = MAX\left(\frac{\overline{\dot{\varepsilon}_{xx}}+\overline{\dot{\varepsilon}_{yy}}}{2} + \sqrt{\left(\frac{\overline{\dot{\varepsilon}_{xx}}-\overline{\dot{\varepsilon}_{yy}}}{2}\right)^2 + \overline{\dot{\varepsilon}_{xy}}^2}; \frac{\overline{\dot{\varepsilon}_{xx}}+\overline{\dot{\varepsilon}_{yy}}}{2} - \sqrt{\left(\frac{\overline{\dot{\varepsilon}_{xx}}-\overline{\dot{\varepsilon}_{yy}}}{2}\right)^2 + \overline{\dot{\varepsilon}_{xy}}^2}\right) \tag{4}$$

$$\overline{\dot{\varepsilon}_{min}} = MIN\left(\frac{\overline{\dot{\varepsilon}_{xx}}+\overline{\dot{\varepsilon}_{yy}}}{2} + \sqrt{\left(\frac{\overline{\dot{\varepsilon}_{xx}}-\overline{\dot{\varepsilon}_{yy}}}{2}\right)^2 + \overline{\dot{\varepsilon}_{xy}}^2}; \frac{\overline{\dot{\varepsilon}_{xx}}+\overline{\dot{\varepsilon}_{yy}}}{2} - \sqrt{\left(\frac{\overline{\dot{\varepsilon}_{xx}}-\overline{\dot{\varepsilon}_{yy}}}{2}\right)^2 + \overline{\dot{\varepsilon}_{xy}}^2}\right) \tag{5}$$

As underlined by previous authors (e.g. Ward (1998); Pancha (2006)), the conversion of surface strain to a scalar moment rate bears large uncertainties and there is no unique method. We use three different equations for calculating the moment rate,

to propagate this uncertainty up to the final moment estimate:

– The Working Group on California Earthquake Probabilities (1995), uses the difference between the principal strain rates:

$$\dot{M}_{0G} = 2\mu AH(\overline{\dot{\varepsilon}_{max}} - \overline{\dot{\varepsilon}_{min}}) \tag{6}$$

– Savage and Simpson (1997), propose that the scalar moment rate is at least as large as:

$$\dot{M}_{0G} = 2\mu AH MAX(|\overline{\dot{\varepsilon}_{max}}|, |\overline{\dot{\varepsilon}_{min}}|, |\overline{\dot{\varepsilon}_{max}} - \overline{\dot{\varepsilon}_{min}}|) \tag{7}$$

– Stevens and Avouac (2021), uses the second invariant, which reflects the magnitude of the total strain rate:

$$\dot{M}_{0G} = C_g\mu AH\sqrt{\overline{\dot{\varepsilon}_{xx}}^2 + \overline{\dot{\varepsilon}_{yy}}^2 + 2\overline{\dot{\varepsilon}_{xy}}^2} \tag{8}$$



With $\mu$ the shear modulus and $H$ the seismogenic thickness. $C_g$ is a geometric coefficient, it depends on the orientation and dip angle ($\delta$) of the fault plane accommodating the strain. Following Stevens and Avouac (2021), for dip-slip faults with

uniaxial compression, $C_g = 1/[sin(\delta).cos(\delta)]$. A dip of 45° corresponds to a geometric coefficient equal to 2, which is the value assumed by Working Group on California Earthquake Probabilities (1995) and Savage and Simpson (1997), as well as in a large part of the literature (e.g. Ward (1998); Jenny et al. (2004); Bird and Liu (2007); D'Agostino (2014)). In their study focused on the Himalayan region, Stevens and Avouac (2021) consider two values, corresponding to dips between 45° ($C_g = 2$) and 15° ($C_g = 4$), to account for the low-angle thrust faults in the region. Here we consider two values 2 and 2.6, which is the

range corresponding to a dip between 25° and 65°.

     The uncertainty on the shear modulus is also taken into account, including two alternative values $3.3 * 10^{10} N.m^{-2}$ and $3.0 * 10^{10} N.m^{-2}$ (e.g. Dziewonski and Anderson (1981)). Whereas for the seismogenic thickness ($H$ in Equations 6 to 8), as there is considerable uncertainty, we use three alternative values (5, 10, and 15 km). This seismogenic (or elastic) thickness is the average thickness over which a region's principal faults store and release seismic energy (Ward (1998)). The thickness

considered in the literature usually varies between 10 and 15km. Pancha (2006) used a fixed seismogenic thickness of 15km throughout the Basin and Range region in Western US. D'Agostino (2014) applied a thickness of $10 \pm 2.5$km throughout the Apennines in Italy, whereas Stevens and Avouac (2021) considered 15km in the India-Asia collision zone. In a study extending over Canada, Ojo et al. (2021) constrained the crustal thickness inferred from ambient seismic noise tomography.

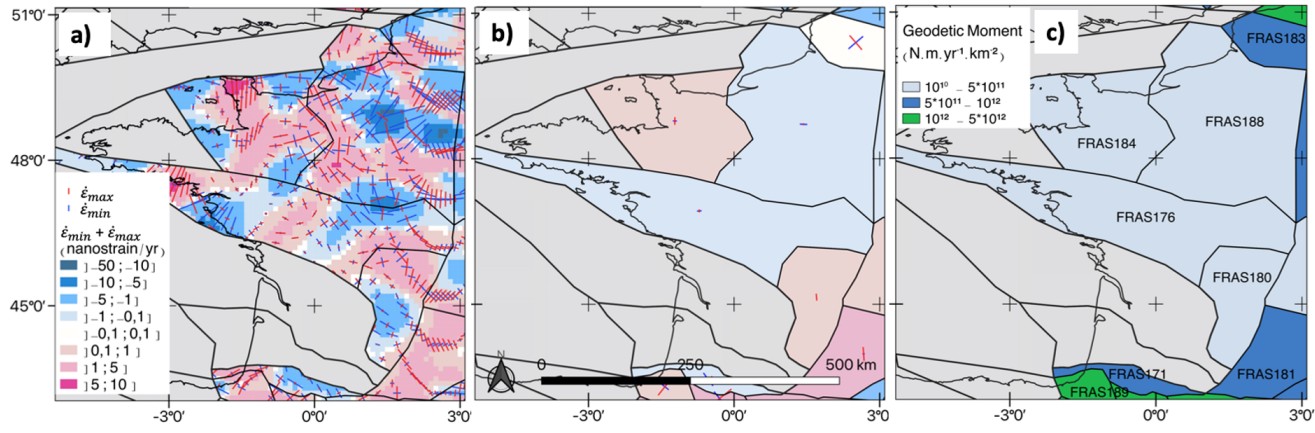

**Figure 3.** Scalar geodetic moment computed from a mean strain tensor, example for the source zones in Northwestern France. (a) Horizontal strain rate tensor from Piña-Valdés et al. (2022) best model, for each grid cell : principal components of the strain rate tensor ($\dot{\varepsilon}_{min}$ in red; $\dot{\varepsilon}_{max}$ in blue) and deformation style ($\dot{\varepsilon}_{min} + \dot{\varepsilon}_{max}$, red : extension, blue : compression). (b) Mean strain rate tensor per source zone, mean principal components in the source zone ($\bar{\bar{\varepsilon}}_{min}$ and $\bar{\bar{\varepsilon}}_{max}$) (Equation 4, 5). (c) One estimate for the geodetic moment rate within the source zone, using the strain rate best model of Piña-Valdés et al. (2022) and considering a depth of 10km, a shear modulus of $\mu = 3.3 * 10^{10}$ $N.m^{-2}$, the equation from Savage and Simpson (1997), and a geometric coefficient $C_g$ equal to 2. Acronyms of ESHM20 area source zones are indicated.





### 1.2.3 A geodetic moment rate distribution per area source zone

The aim is to obtain a distribution for the moment rate within an area source of ESHM20, that is representative of the uncertainties. Figure 4 displays the exploration tree set up to combine 12 different preprocessing parameters to filter the stations of the GNSS velocity fields (choice on the class and on the radius applied to identify outliers), with 12 different regularizations of the velocity fields inversion to determine strain rates (choice of the distance and spatial weighting scheme, choice of the weighting threshold) and with finally 36 different parametrizations to calculate the moment rate from the strain rates. For a given source

zone area, we obtain 5184 alternative moment rate estimates ($12 * 12 * 36$). Figure 4b displays the distribution obtained for the area source zone hosting Paris in France. The variability of the moment rate is significant, the value corresponding to the percentile $84^{th}$ is three times larger than the value corresponding to the percentile $16^{th}$.

To understand which parameters, or decision, control the most the overall variability on the geodetic moment, different parts of the tree are explored (Figure 5, see Mariniere et al. (2021)). The analysis is led in 3 sample areas source zones

characterized by different seismic activity: southern Brittany in France, located in an intracontinental region and characterized by a low seismic activity, a large source zone in Fennoscandia in a very low seismicity region, and northern Tuscany in Italy, a moderate seismic region (see Fig. 6 for locations). For every parameter choice, the entire tree is explored keeping fixed the other parameters, then from the distribution obtained the mean as well as the percentiles $16^{th}$ and $84^{th}$ are estimated. For example, exploring separately the alternative branches corresponding to the three different selections of GPS stations yields

3 distributions, made of 1728 moment estimates each (in green). Exploring separately the branches based on the 2 alternative spatial weighting schemes yields 2 alternative distributions, made of 2592 moment estimates each (in pink). The larger is the dispersion obtained between the alternative mean values of the distributions, the larger is the contribution of this parameter uncertainty to the overall moment variability.

The results show that the uncertainty on the seismogenic thickness controls the overall moment variability, for all area source

zones. The geodetic moment exhibits a linear variation with both the seismogenic thickness and the shear modulus. Except for the shear modulus for which a limited range of values is explored, all other parameters uncertainties also contribute to the overall variability. It is interesting to note that the exact selection of GNSS stations has an influence on the moment rate estimates only in low seismicity areas (Fennoscandia and Southern Brittany), but no impact in the moderate to high seismicity areas (such as northern Tuscany). This phenomenon can be attributed to the dominance of strong strains in high-deformation

zones, where even lower-quality stations provide accurate measurements at a first-order approximation. Conversely, in low-deformation areas, the measured signal is close to the noise level. Consequently, the exclusion or inclusion of one or more stations has a substantial impact.





**Figure 4.** Determination of a distribution for the moment rate per area source zone, taking into account the uncertainties on the different steps. (a) Exploration tree to account for the uncertainty on the exact set of GNSS stations used, on the technique applied to infer strain rates from the geodetic velocities, and on the parameters used to calculate the moment rate within an area source. (b) distribution of the geodetic moment rate estimates (5184 values) obtained for the example source zone Parisian Basin in France (FRAS188 in ESHM20), mean value (red) and percentiles $16^{th}$ and $84^{th}$ (blue). (c) Three alternative distributions for the moment rate estimates, depending on the choice of the seismogenic depth, example source zone Parisian Basin in France

## 1.3 Is ESHM20 earthquake rate forecast consistent with the tectonic loading measured by geodesy?

Our aim is to compare the moment rate corresponding to the long-term ESHM20 earthquake rate forecast with the geodetic
moment rate. We acknowledge that the comparison between deformation measurements performed over a few decades and a seismogenic source model for a regional seismic hazard assessment must be done with caution. The ESHM20 earthquake rate forecast relies on earthquake catalogs extending over several centuries. The recurrence model is in general anchored on





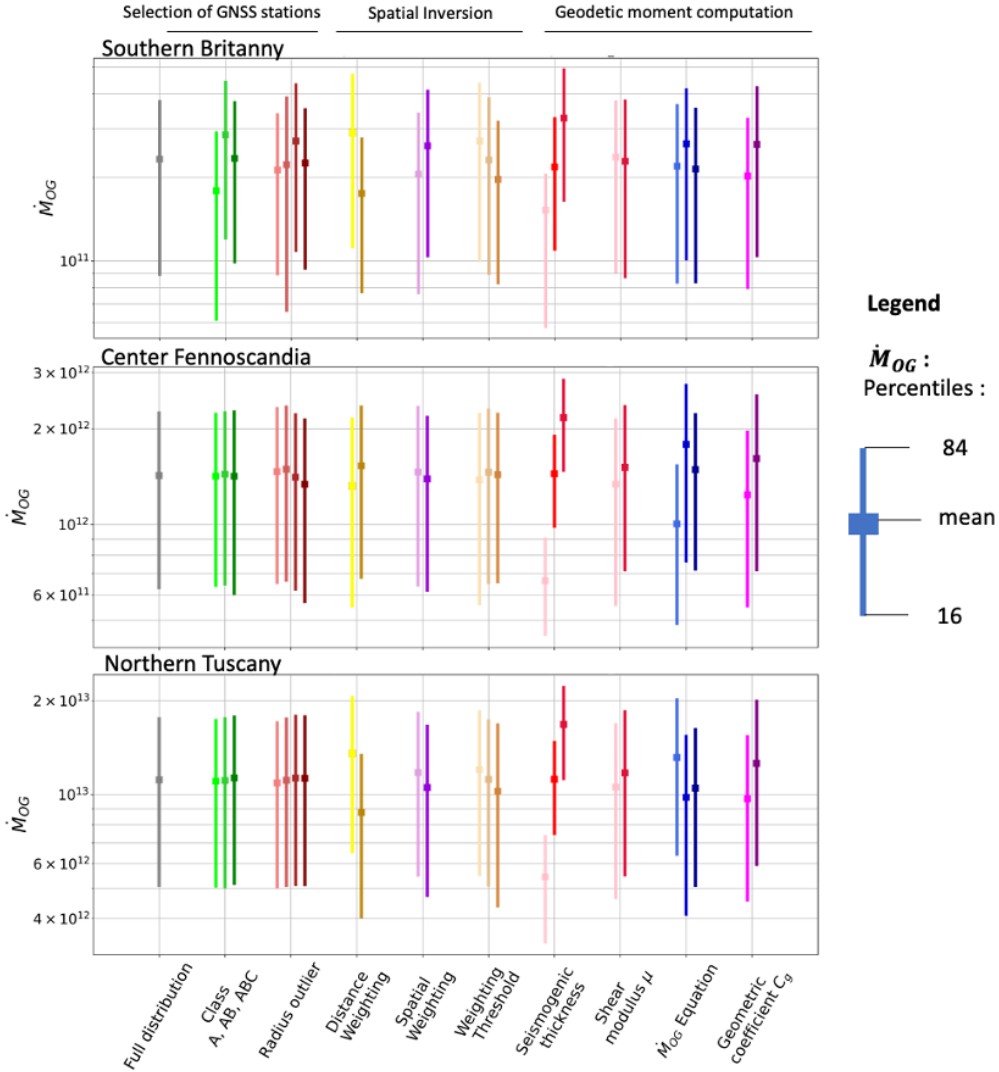

**Figure 5.** Distribution for the geodetic moment rate ($\dot{M}_{0G}$) and identification of controlling parameters, in 3 example source zones: southern Brittany (FRAS176), Fennoscandia (SEAS410), and Northern Tuscany in Italy (ITAS335), see location in Figure 6). Mean value (square), as well as $16^{th}$ and $84^{th}$ percentiles (vertical bar). "Full": full exploration of the tree (5184 branches' combination and moment values). "Class A, AB, ABC": 3 different sets of GNSS stations, according to quality (1728 values each). "Radius outlier": choice of the spatial radius for discarding outliers (50, 100, 150, 200 km, from salmon to dark red, 1296 values each). "Distance weighting scheme": choice of the decay function used for interpolation, whether Gaussian or Quadratic (2592 values each). "Spatial weighting scheme": choice of the method for spatial inversion, whether Azimuth or Voronoi. "Weighting Threshold": Choice of the threshold value on the distance weighting function (6, 12, 24, increasing smoothing, beige to brown, 1728 values each). "Seismogenic depth": elastic depth (5, 10 and 15 km, pink to red, 1728 values each). "$\mu$": choice of shear modulus value ($3.3 * 10^{10}$ $N.m$ (pink), $3 * 10^{10}$ $N.m$ (red)). "$\dot{M}_{0G}$ equation": choice of the geodetic moment equation, see the text. "$C_g$". Choice of the geometric coefficient parameter, 2 (pink) or 2.6 (purple)





**Figure 6.** Area source zones mentioned throughout the manuscript. In green: example source zones in section 1.2.3: FRAS176 in Southern Brittany in France, SEAS410 in Fennoscandia, ITAS335 in northern Italy, as well as GRAS257 in Greece in section 1.3.2. In pink : the eight source zones where the geodetic moment estimates is much lower than the seismic moment estimates (section 1.3.3 and Fig. 10). The grey dashed line represents the zones considered affected by the Scandinavian GIA, including those intersecting this line and those located to the north. The selection is based on the vertical velocity signal (Piña-Valdés et al., 2022) and includes 18 zones.





the observed earthquake rates extrapolated up to magnitudes that correspond to the largest possible events in the area sources.
The model thus relies on both recent observations (instrumental eq. catalogue) and past historical seismicity well as on a wider
analysis of the seismogenic potential of the area. The earthquake rate forecast model also includes our current knowledge about
active faults (fault traces, segmentation, extension at depth). Geodetic information has been used in some cases for estimating
the deformation accumulating along these faults (Basili et al. (2023)). The strain model thus is not strictly independent from
the source model, however GNSS velocities have not been directly used to build the ESHM20 source model. The strain rate
model can be used to test the ESHM20 source model and evaluate how realistic the model is.

**1.3.1    Correlation between geodetic and seismic moment rates at the scale of Europe**

The geodetic moment rate quantifies the ground surface deformation, that encompasses both seismic and aseismic processes.
The mean moment estimates obtained in every area source zone are displayed in Figure 7. Overall, geodetic moment rates
appear larger or equal to seismic moment rates, similarly to the findings of many previous studies (e.g. Ward (1998); Jenny
et al. (2004); Mazzotti et al. (2011)). Largest geodetic and seismic rates are found in Greece, in Italy and in the Balkans. The
distribution in space of the geodetic moment rate is much more smoothed than the seismic moment rate. One explanation could
be that the deformation measured by geodesy is more representative of long-term processes than the earthquake catalogs.
If earthquake catalogs of much longer time windows were available (e.g. 100,000 years), would the spatial distribution of
the seismic moment rates be more alike the spatial distribution of the geodetic moment rates? Another explanation could be
that the geodetic moment rate has a lower resolution in space than the seismic moment rate inferred from the modeling of
earthquake recurrence. Indeed, because of the smoothing procedure applied to derive the strain rates, the geodetic moment is
strongly correlated spatially. Besides, we observe that in low-seismicity regions, geodetic moment rates stay between $\sim 10^{11}$
and $10^{12} N.m.yr^{-1}.km^{-2}$ (in blue and green, in mainland Spain and France, northern Europe and Fennoscandia) whereas the
seismic moment rates go down to much lower values.

Figure 8 shows a comparison between geodetic and seismic mean moment in Europe at the scale of the ESHM20 source
zones. It demonstrates a remarkable linear correlation between the geodetic and seismic moment rates above $\sim 2.10^{11} N.m.yr^{-1}.km^{-2}$.
In general, in the most active regions in Southern Europe, the geodetic moment rates are well correlated with the seis-
mic moment rates. On the contrary, in the less active regions in northern Europe, above $\sim 50°$ latitude, the geodetic mo-
ment appears completely decorrelated from the seismic moment. Seismic moment rates decrease to levels as low as $10^9 - 10^{10} N.m.yr^{-1}.km^{-2}$, whereas geodetic moment rates reach a plateau around $10^{12} N.m.yr^{-1}.km^{-2}$. The deformation mea-
sured in Fennoscandia and surrounding regions might be mostly related to the post-glacial rebound and only a very small part
of it might be tectonic deformation (Keiding et al. (2015); Craig et al. (2016)).





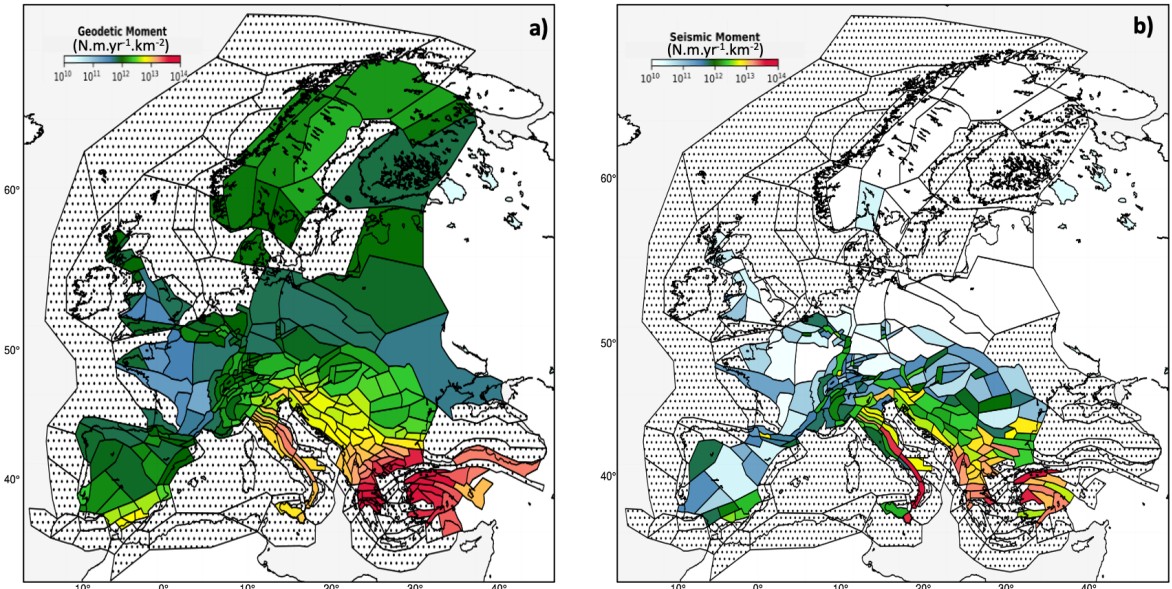

**Figure 7.** Mean geodetic and seismic moment rates within the ESHM20 area source zones. (a) Mean geodetic moment ($\dot{M}_{0G}$) based on the strain rates, mean of the distribution obtained by exploring uncertainties; (b) Mean seismic moment ($\dot{M}_{0S}$) estimated from the ESHM20 source model logic tree. Area sources with more than 35 % of surface off-shore, or where the density of GNSS station is too low ($\leq 1$ station per $100000 \ km^2$) are discarded.

### 1.3.2 Comparison of the moment rates distributions

Rather than comparing only mean values of distributions, the comparison of the full distributions can be more instructive as the uncertainties are accounted for. For a given area source zone, the distribution for the geodetic moment rate relies on 5184

alternative values (see Section 1.2.3), whereas the distribution for the seismic moment is built from the 21 alternative branches in the ESHM20 source model logic tree, taking into account the weights associated to each branch (for each branch, the number of values is duplicated according to its weight in the logic tree, to reach a total number of value close to 5184).

In Fennoscandia, the geodetic moment estimates are on average one hundred to three hundred higher than the seismic moment estimates ($log_{10}(\dot{M}_{0S}/\dot{M}_{0G})$ varies between -2 and -2.5, in red in Fig. 8). The uncertainty on the geodetic moment is

large, but still there is no overlap between the two distributions (example source zone SEAS410, Fig. 9). In most area sources below latitude 52°, geodetic moment estimates are larger or equal to seismic moment rates, up to five times higher on average than seismic moment rates ($log_{10}(\dot{M}_{0S}/\dot{M}_{0G})$ varies between 0 and -0.7, in green and yellow in Fig. 8). In some area sources such as GRAS257 in Greece, the mean geodetic moment rate results five times higher than the mean seismic moment rate and their distributions only partially overlap. In other source zones, such as FRAS176 in France or ITAS335 in Italy, the seismic

and geodetic distributions are very consistent.




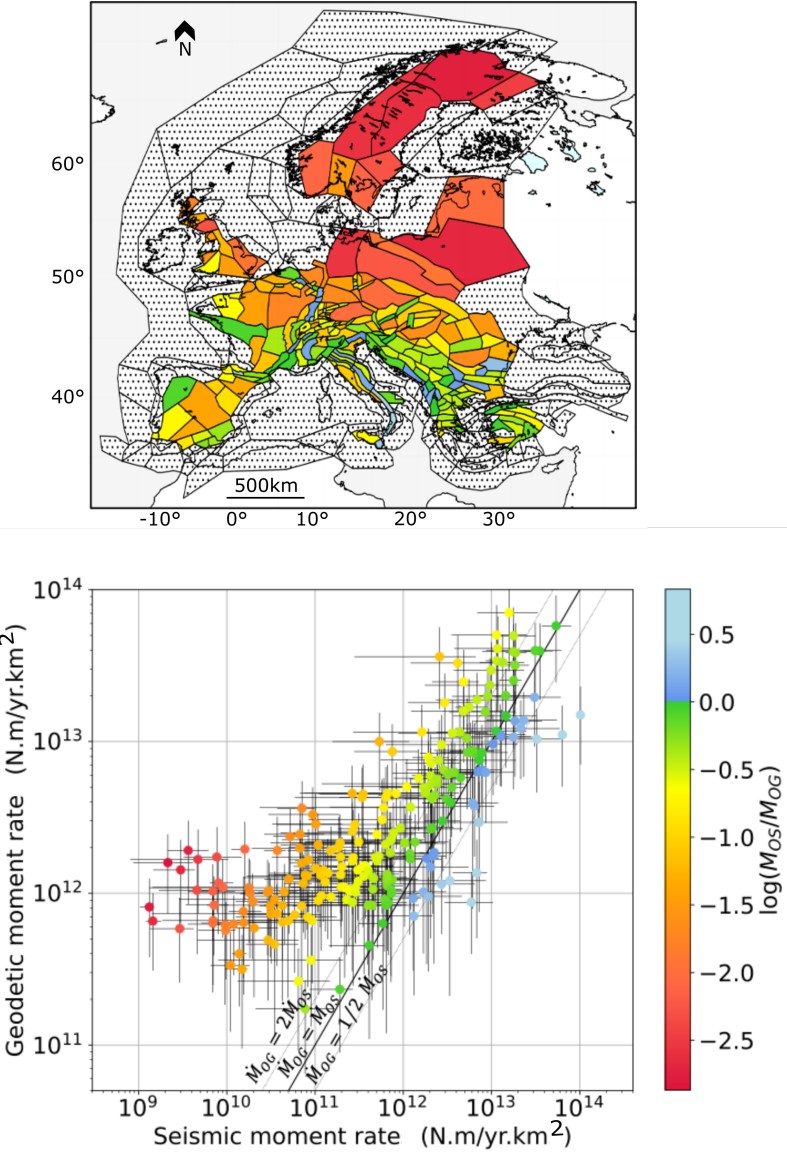

**Figure 8.** Comparison between geodetic and seismic moment in Europe at the scale of the ESHM20 source zones : mean $\dot{M}_{0G}$ versus mean $\dot{M}_{0S}$ at the scale of the source zone (uncertainty range $16^{th}$ to $84^{th}$ percentile indicated).





We quantify the overlap between the geodetic and the seismic distributions, for all area sources (Fig. 10). As the distributions are in most cases unimodal, the overlap between the distributions is usually increasing with closer mean moment values. In the most seismically active regions in Europe, i.e. in Greece, Italy, the Balkans, as well as in some parts of France and Switzerland, the seismic and geodetic moment estimates are rather consistent (overlap between 35 and 80%, in blue); whereas elsewhere 260 the fit is quite poor (overlap lower than 30%, in red).

**Figure 9.** Comparison of seismic and geodetic moment rate distributions, for 4 example source zone in Fennoscandia (SEAS410), Greece (GRAS257), France (FRAS176) and Italy (ITAS335), source zones in Fig. 6. The overlap is computed as : 0 if $MIN(\dot{M}_{0G}) > MAX(\dot{M}_{0S})$ or if $MIN(\dot{M}_{0S}) > MAX(\dot{M}_{0G})$ and as $\sum_{bins} MIN(\dot{M}_{0G}, \dot{M}_{0S})/$ number element($\dot{M}_{0G}$).



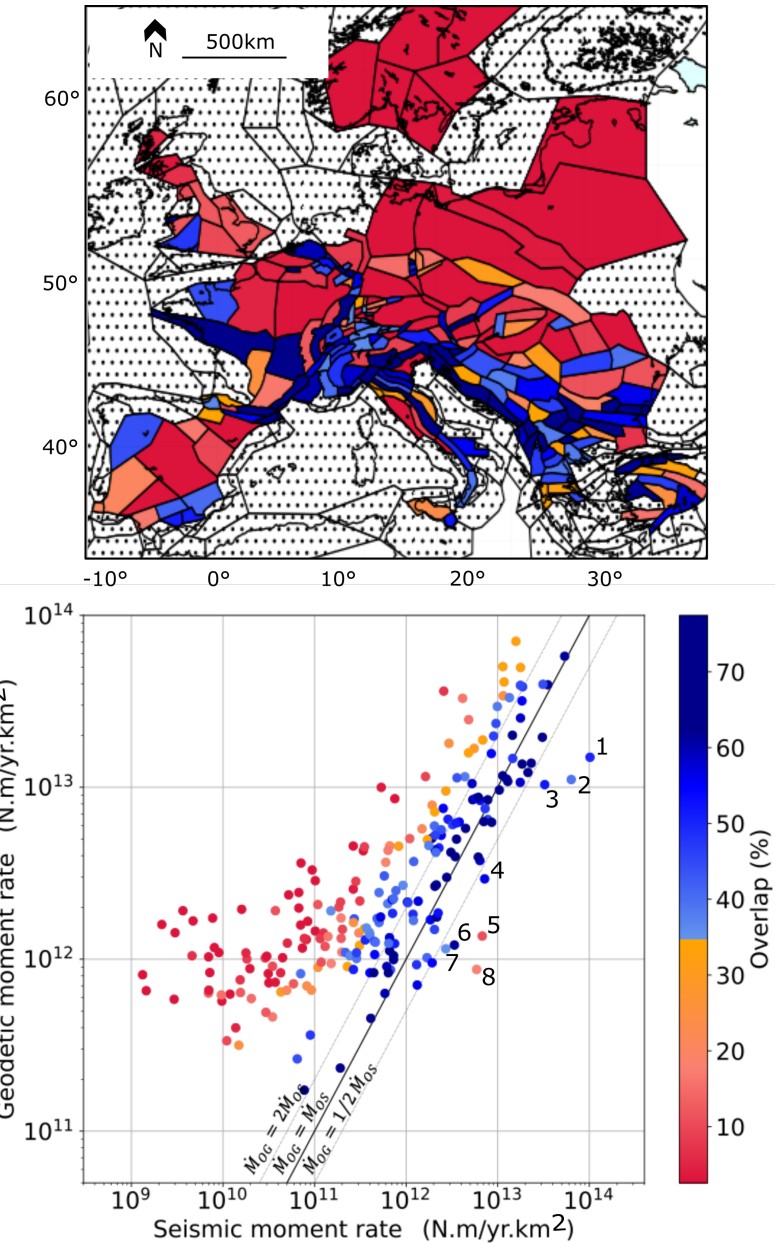

**Figure 10.** Comparison between geodetic and seismic moment rate mean estimates, within the ESHM20 shallow area source zones (227 source zones considered), estimates for the overlap between the seismic and geodetic distributions. Shallow area source zones where the geodetic moment rate is much lower than the seismic moment rate : 1 : ITAS308 , 2 : ITAS331 ; 3 : ITAS339 , 4 : BGAS043 , 5: FRAS164, 6: DEAS113, 7: DEAS109, 8: CHAS071 (see the text and Fig. 6).





As the size of some source zones is too small for the comparison to be meaningful (see Section 1.3.3), we also perform the comparison at the scale of the macrozones. Macrozones are used as a spatial proxy in the building of the ESHM20 seismogenic sources, here we use the macrozones named 'TECTO' which corresponds to the seismo-tectonic layer of the region. (Fig. 2). Danciu et al. (2021) used these macrozones to evaluate the b-value for the smoothed seismicity model, for the underlying crustal active faults and/or to constrain GR parameters of those area sources without sufficient number of earthquakes. As expected, at the scale of Europe, the correlation between the seismic and geodetic moment rates is slightly improved when considering the macrozones, that cover a much larger spatial region than the individual area source zones (right column in Fig. 2). Performing the comparison at the scale of the macrozones, a rather good fit is obtained for the whole Euro-Mediterranean region, except in Spain (Fig. 11).




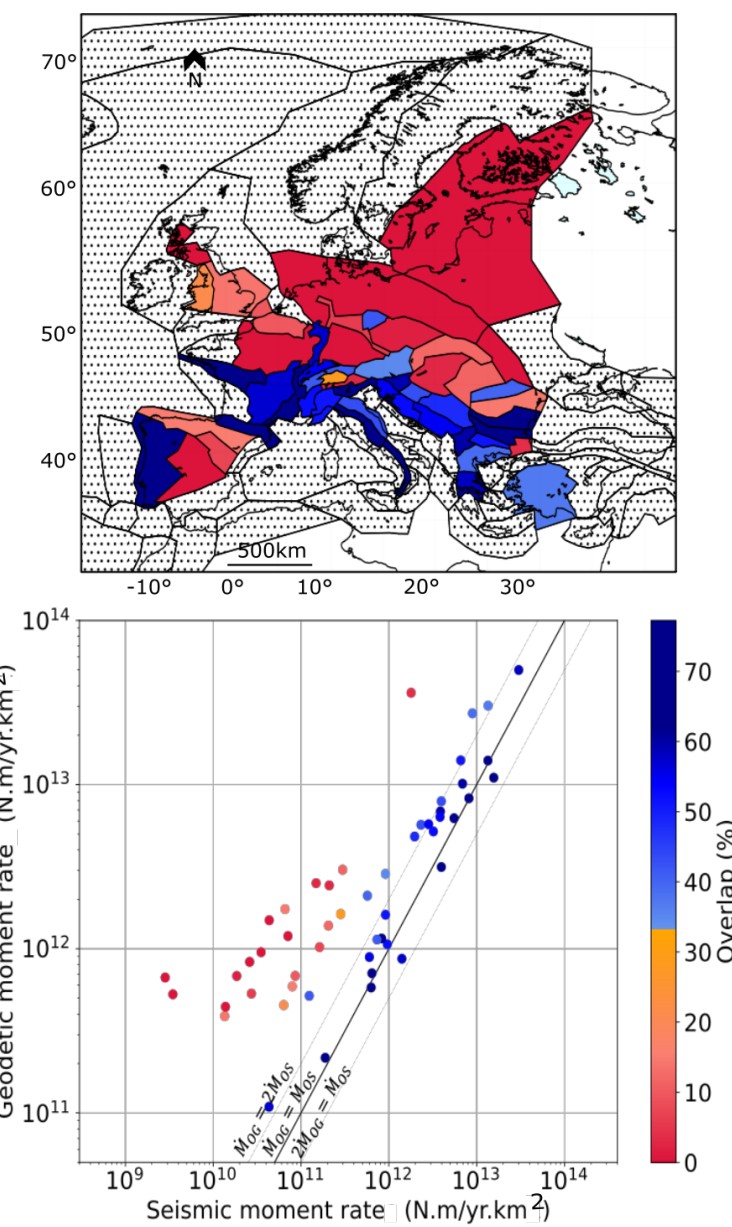

**Figure 11.** Comparison between geodetic and seismic moment rate mean estimates, within the ESHM20 'TECTO' macrozones (51 macro-zones considered), the overlap between the seismic and geodetic distributions is indicated





### 1.3.3 A closer look to the area sources with a seismic moment much higher than the geodetic moment

Let's have a closer look at the eight area sources where the geodetic moment rate results on average significantly lower than the seismic moment rate (data points that are below the straight line $\dot{M}_{0G} = \dot{M}_{0S}/2$ in Fig. 8).

These area sources fall in three categories:

1. Small size area, below the resolution of the geodetic signal (FRAS164, CHAS071, DEAS113, DEAS109). The source zone FRAS164 in Western Pyrenees is a small area with high seismic activity in comparison with the neighboring area zones. The geodetic signal has a too large spatial wavelength to capture these rapid spatial changes. The source zones CHAS071, DEAS113 and DEAS109 are not as active, but they are also small size area sources. In these cases, the difference between the geodetic and the seismic moment estimates is expected.

2. Areas with poorly constrained earthquake recurrence model, due to insufficient number of events for statistical fitting (i.e., ITAS339, or BGAS043). In those areas, the resulting ESHM20 recurrence model fits the observed rates in the upper magnitude range, but predicts larger seismic rates than what has been observed in the past in the moderate magnitude range. In these areas, there are too few data to constrain the model and the b-value is inferred from the larger macrozone Danciu et al. (2021).

3. Areas where unusual earthquake recurrence models have been proposed to account for two different slopes observed in the Gutenberg-Richter model (area model, ITAS331, ITAS308). In both area sources, the slope of the recurrence model in the upper magnitude range (mostly historical period) is lower than the slope in the moderate magnitude range (mostly instrumental period). This is quite unusual and does not fit a Gutenberg-Richter model characterized by a unique b-value. It is interesting to note that the fault model branches overall provide a moment range that is consistent with the geodetic moment range, whereas the area model branches lead to much higher moment estimates.





### 1.3.4   The consistency between the geodetic and seismic moments depends on the activity level, the spatial scale, and the source model

The Figure 12 provides an overall view on the comparison between geodetic and seismic moment estimates at the scale of Europe, and how this comparison varies when subsets of areas or branches of the ESHM20 source model logic tree are selected. Distributions for the ratio $log_{10}(\dot{M}_{0S}/\dot{M}_{0G})$ are characterised by mean values, as well as percentiles 16th and 84th (boxplots). When $log_{10}(\dot{M}_{0S}/\dot{M}_{0G})$ tends towards 0, $\dot{M}_{0S}$ and $\dot{M}_{0G}$ get closer. Area source zones are also grouped according to their level of geodetic moment estimates (dark green, light green, red), showing that, when all area source zones are considered, the consistency improves with increasing deformation rate.

In area source zones affected by the glacial isostatic adjustment (18 zones, selected based on their vertical velocity signal (Piña-Valdés et al., 2022)), both the geodetic and the seismic moments are low, but the moment estimate based on modeled earthquake recurrence distributions is two orders of magnitude lower than the geodetic moment. This suggests that the deformation processes involved are mostly aseismic, which is compatible with the processes involved in glacial isostatic adjustment (GIA). GIA generates a viscous asthenospheric flow and a large scale flexure of the overlying elastic lithosphere that results in rather large wavelength deformation (i.e. the strain is distributed over a large area). It should also be noted that the post-glacial rebound is a phenomenon that is not representative of the long term (a few Myrs) tectonics, but that it is a transient mechanisms that started after the last glacial maximum, ~20,000 years ago. The cumulative deformation associated with the glacial isostatic adjustment may therefore not reach locally the strength threshold of the Fennoscandian lithosphere, although this may vary spatially depending on the elastic thickness of the crust (Perez-Gussinye et al. (2003)). In those areas, the surface deformation measured by geodesy is therefore not a good proxy for the seismic activity and can not be used directly to constrain seismogenic source models.

Considering area sources with the best constrained recurrence models (at least 30 events used, see Danciu et al. (2021)), the consistency between both moment rate estimates is strongly improved. In general, the zones with the largest number of events falling inside periods of completeness are the zones with highest seismic activity.

The compatibility between geodetic and seismic moment rates varies depending on the branch of the ESHM20 seismogenic source model logic tree ( 2) , 3) and 4) in figure 12), from which the zones affected by GIA have been removed. Considering the smoothed seismicity and fault model branch, the seismic moment rates are overall less consistent with the geodetic moment rates than in the area model. The comparison is done at the scale of the area sources. We group the area zones that include faults on one side, and the area zones that do not include any fault on the other side. We observe that the geodetic and seismic moment rates are much more consistent in the first group, the faults have mostly been characterized in the most seismically active parts of Europe. Considering the area branch of ESHM20 model ((3) and (4) in figure 12), we check if the fit between geodetic and seismic moment rates varies with the model selected to extrapolate earthquake rates in the upper magnitude range. The fit results slightly better using the classical Anderson and Luco (1983) form 2 than the Pareto distribution. This result is expected, as the Pareto distribution implies a stronger decrease of seismic rates in the upper magnitude range with respect to Anderson and Luco distribution (therefore a lower seismic moment rate). We also perform the comparison at the scale of the



macrozones (right column). The correlation between the seismic and geodetic moment rates is slightly improved, mean values

of distributions tend to be closer to 0, macrozones cover a much larger spatial region than the individual area source zones.

The improved consistency between seismic and geodetic moments with increasing deformation rates is also observable in figure 13 , that displays the ratio $log_{10}(\dot{M}_{0S}/\dot{M}_{0G})$ as a function of the number of earthquakes used to constrain the earthquake recurrence model in the corresponding area source. The size of the symbol increases with the density of faults in the zone, and the color reflects the level of geodetic moment (as in figure 12). This figure shows mean values only. Zones with highest

geodetic moment rates exhibit better consistency between $\dot{M}_{0S}$ and $\dot{M}_{0G}$ (all zones with $\dot{M}_{0G} \geq 10^{13} N.m.yr^{-1}.km^{-2}$, in red, fall within the $log_{10}(\dot{M}_{0S}/\dot{M}_{0G})$ interval of -1 to 0.5). Zones with $log_{10}(\dot{M}_{0S}/\dot{M}_{0G})$ below -1 are mostly characterized by low strain rates ($\dot{M}_{0G} \leq 3.10^{12} N.m.yr^{-1}.km^{-2}$) and include areas affected by GIA. In areas where the recurrence model was constrained with at least 50 events (most active areas in Europe), the $log_{10}(\dot{M}_{0S}/\dot{M}_{0G})$ ratios fall mostly between -0.5 and 0.5 (i.e. the ratio between seismic and geodetic moments rates are within a factor 1/3 to 3).

The Figure 13 also highlights the impact of active fault density on the consistency between geodetic moment rates. Active fault density for each zone is defined as the length of faults with a slip rate exceeding 0.1 mm/yr, divided by the zone's area (expressed in $km^{-1}$). Zones with a density above $4*10^{-5} km^{-1}$ display a $log_{10}(\dot{M}_{0S}/\dot{M}_{0G})$ between -1 and 1, with the majority falling within -0.5 and 0.5. Higher active fault density tends to enhance compatibility between $\dot{M}_{0S}$ and $\dot{M}_{0G}$. We have not checked the respective contribution of the faults and the smoothed seismicity to the area moment rate. This

reasoning is based on the hypothesis that the faults are controlling the area moment rate, which might not be true, especially in low seismicity areas. Similar results are observed in Figure 12 (part (2)) when we consider only the ESHM20 model branch based on smoothed seismicity and faults (although this branch exhibits seismic and geodetic moment rate estimates that are overall less consistent than is other branches of ESHM20) : We observe that the geodetic and seismic moment rates are much better correlated in the area zones that include faults than in area zones that do not include any fault in the model. This might be

correlated to the activity of the area, since the faults are easier to map and characterize in the seismically active region. However, in figure 12 (2) and figure 13, we can observe that in zones with lower strain ($\dot{M}_{0G} \leq 3*10^{12} N.m.yr^{-1}.km^{-2}$), increasing the fault density or the number of earthquakes used to constrain the earthquake recurrence model brings the $log_{10}(\dot{M}_{0S}/\dot{M}_{0G})$ closer to 0. Zones with $log_{10}(\dot{M}_{0S}/\dot{M}_{0G})$ below -1 are all characterized by a minimum fault density ($\leq 2*10^{-5} km^{-1}$) and less than 30 earthquakes used to constrain the earthquake recurrence model. This suggests that the inclusion of active faults may

strengthen the earthquake recurrence model in areas that are characterized by both a slow deformation rate and rare seismic events. As a corollary, this may indicate that, in areas where enough active faults are identified and mapped by geologists, the geodetic moment rate may be used as a proxy for the long term tectonic loading, even in areas where a limited number of earthquakes is available to constrain the recurrence model.



**Figure 12. Consistency between $\dot{M}_{0S}$ and $\dot{M}_{0G}$ at the area source zone scale (left) and at the macrozone scale (right)**. Whisker plots indicate mean, as well as percentiles 16th and 84th, 5th and 95th. Black circles: individual ratio per source zone. Area sources are also grouped per increasing geodetic moment level: dark green $\dot{M}_{0G} > 3 * 10^{12}$ $N.m.yr^{-1}.km^{-2}$, green $3 * 10^{12} < \dot{M}_{0G} < 10^{13}$ $N.m.yr^{-1}.km^{-2}$, and red $\dot{M}_{0G} > 10^{13}$ $Nm.yr^{-1}.km^{-2}$. In subsets (2), (3) and (4), the 18 zones affected by the Fennoscandian Glacial Isostatic Adjustment (GIA) have been discarded (see figure 6). 227 area source zones considered in total, including 18 considered affected by GIA, 57 well-constrained recurrence model, and 110 are including faults.



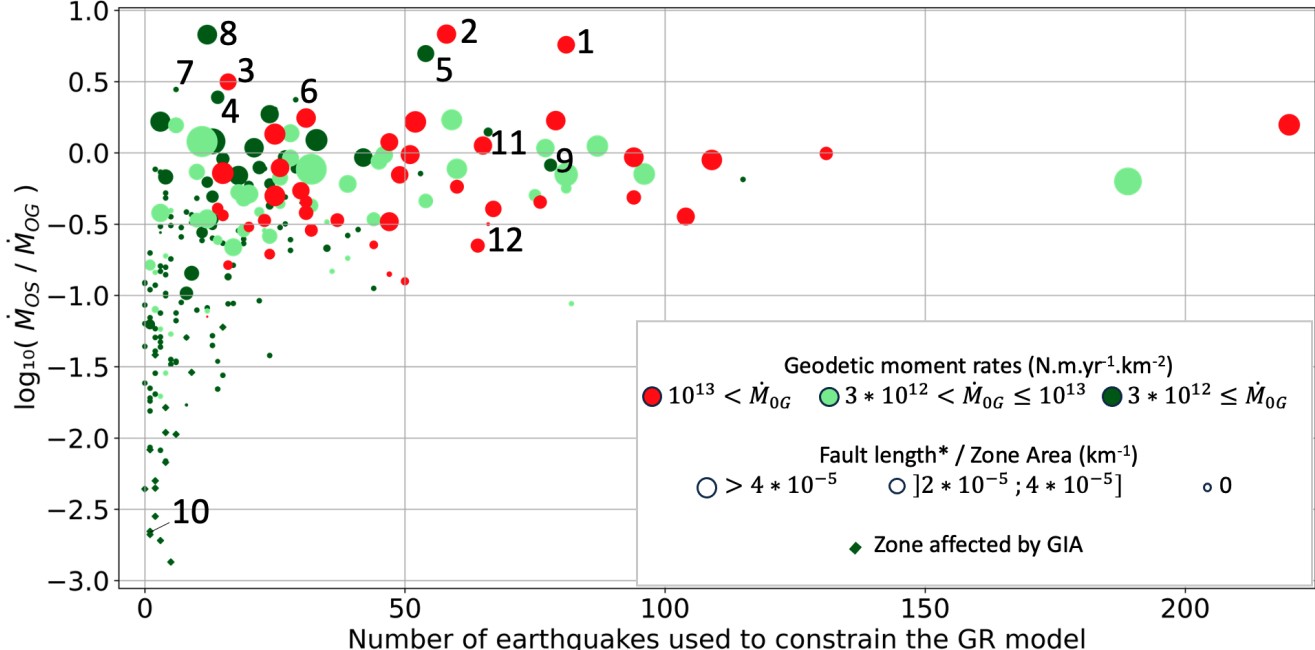

**Figure 13.** Mean $log_{10}(\dot{M}_{0S}/\dot{M}_{0G})$ for all source zones in Europe, as a function of the number of earthquakes used to constrain the earthquake recurrence model ($M_W \geq 3.5$). The color represents the mean geodetic moment of the source zone area, and the size of the symbol is proportional to the density of the faults, which slip rates is higher than 0.1mm/yr (*), in the ESHM20 fault model. Compatibility between geodetic and seismic moment rates increases with the geodetic moment rates, the number of earthquakes used to constrain the earthquake recurrence model, and the fault density. Shallow area source zones where the geodetic moment rate is much lower than the seismic moment rate : 1 : ITAS308 , 2 : ITAS331 ; 3 : ITAS339 , 4 : BGAS043 , 5 : FRAS164, 6 : DEAS113, 7 : DEAS109, 8 : CHAS071 and example source zones in section 1.2.3 : 9: FRAS176, 10: SEAS410, 11: ITAS335, 12: GRAS257 (see the text and Fig. 6).

## 1.4 Focus in Italy

Figure 14a presents a magnified view of figure 8a in central Italy. As highlighted previously, the central zone (ITAS317) demonstrates a mean seismic moment ($\dot{M}_{0S}$) exceeding the mean geodetic moment ($\dot{M}_{0G}$) ($log_{10}(\dot{M}_{0S}/\dot{M}_{0G}) > 0$). Conversely, the surrounding zones exhibit a geodetic moment significantly higher than the seismic moment ($log_{10}(\dot{M}_{0S}/\dot{M}_{0G}) \sim -1$, overlap $< 20\%$). As a consequence, and as seen at the scale of macrozones (Fig. 11), this discrepancy is reduced at a larger scale because of a spatial smoothing of the signal. In this section, we try to understand the reasons for such peak of seismic release

by examining the seismic moment against the geodetic moment along a profile across the Appenines passing through Rome (profile AB in figure 8a and b).

The figure 14a presents the ratio between the estimated $\dot{M}_{0S}$ and $\dot{M}_{0G}$ at the scale of source zones. The figure 14c provides a view of how these moments are distributed as a function of the distance along the cross-section AB: the average geodetic



($\dot{M}_{0G}$) and seismic ($\dot{M}_{0S}$) moment rates are represented by plain orange bars and empty black bars, respectively. $\dot{M}_{0G}$ exceeds

the mean $\dot{M}_{0S}$ in all source zones (5 to 10 times larger), except in the central source zone, which is the most seismically active and encompasses several faults. In this particular source zone (ITAS317), $\dot{M}_{0S}$ (computed as the weighted mean of all ESHM20 branches, shown by empty black bars) exceeds the geodetic moment estimated from strain rates.

We use the fault and smoothed seismicity model of ESHM20 source model logic tree to compare the seismic moments with the average geodetic moments from the strain rate solutions, evaluated on the same spatial grid. It should however be noted that

the ESHM20's hybrid model forecasts seismic moments that are on average smaller than the models based on area source zones (figure 12, figure 14). The fault and smoothed seismicity model (purple bars) exhibits seismic moments that are systematically lower than the mean inferred from the full ESHM20 source model logic tree.

$\dot{M}_{0S}$ and $\dot{M}_{0G}$ are compared along a profile AB, averaged within spatial bins of 14 km (figure 14b). This analysis at a finer scale reveals that the seismic moment is concentrated on the fault traces (marked with small blue arrows). The geodetic

moment rate exhibits a smoother behavior, and reaches its maximum ($4*10^{13} N.m.yr^{-1}.km^{-2}$) at the level of the eastern fault (similarly to $\dot{M}_{0S}$).

Several propositions can be put forth to explain this observation. Firstly, we may question whether the spatial regularization scheme used during the strain map inversion could lead to a smoothing of the solution. In Italy the density of GPS stations is quite high with an interstation distance of 20km on average, and the network should capture any spatial details larger than

30km in the deformation field. The observed difference in spatial distribution between the seismic moment release and the geodetic moment is most likely real.

Another possible explanation is to invoke the elastic rebound theory. In areas affected by major active faults, the faults accumulate elastic strain that is then released into earthquakes. During the interseismic period, the deformation associated with the loading is usually modeled as a fault that is locked down to a given depth and that is creeping at the loading rate at

greater depths. This generates a surface deformation that has a large spatial wavelength. The deeper the locking, the wider the deformation across the fault. In elastic rebound theory, the slip deficit accumulated during the loading phase is then released into earthquakes located on the fault plane. The elastic rebound theory can therefore explain the observed differences of spatial distribution of $\dot{M}_{0S}$ and $\dot{M}_{0G}$ across the Appenines. In areas where the deformation mechanism is dominated by the seismic cycle on active faults, a proper modelling of the interseismic coupling on the faults would be better adapted Mariniere (2020);

Mariniere et al. (2021).

We can conclude that the scale at which deformation is observed is a crucial criterion for analyzing the compatibility between seismic and geodetic moments. Therefore, in places where source zones enclose fault zones or areas with high seismic activity, the comparison between the geodetic moment and the seismic moment can be meaningful only if led at a large enough spatial scale.



**Figure 14.** Spatial variability of geodetic deformation and seismic release in the central Apennines. a) Mean ratio between $\dot{M}_{0S}$ and $\dot{M}_{0G}$ for area source zones in Central Italy (zoom of Fig 8.a.) b) Mean geodetic moment rate per surface unit inferred from strain rates ; grey dots : earthquakes in the ESHM20 unified earthquake catalog; blue lines : active faults included in the EFSM20. c and d) Geodetic ($\dot{M}_{0G}$) and seismic ($\dot{M}_{0S}$ moment rates per kilometer along the cross-section AB : averaged within the source zones (c), or averaged within bins of 14 km along a 190 km-wide swath profile (thin grey rectangle) (d) Blue arrows : location of the two main faults.



## 1.5 Focus in France


We focus on the area source zones included within France or located on the border with neighboring countries (Figure 15). The distributions of the geodetic and seismic moments determined for each zone are compared in figure 16, with sources zones ordered according to increasing number of earthquakes used to constrain the recurrence models in ESHM20. In regions of very low seismicity, with recurrence model relying on less than 10 events, the distribution of the seismic moment ($\dot{M}_{0S}$) is

systematically much lower than the distribution of geodetic moment ($\dot{M}_{0G}$), with no overlap. In area sources with more than 18 events used to establish the recurrence model, both distributions tend to overlap. There are exceptions, such as FRAS164 in the Western Pyrenees, a small zone with high and episodic seismic activity.

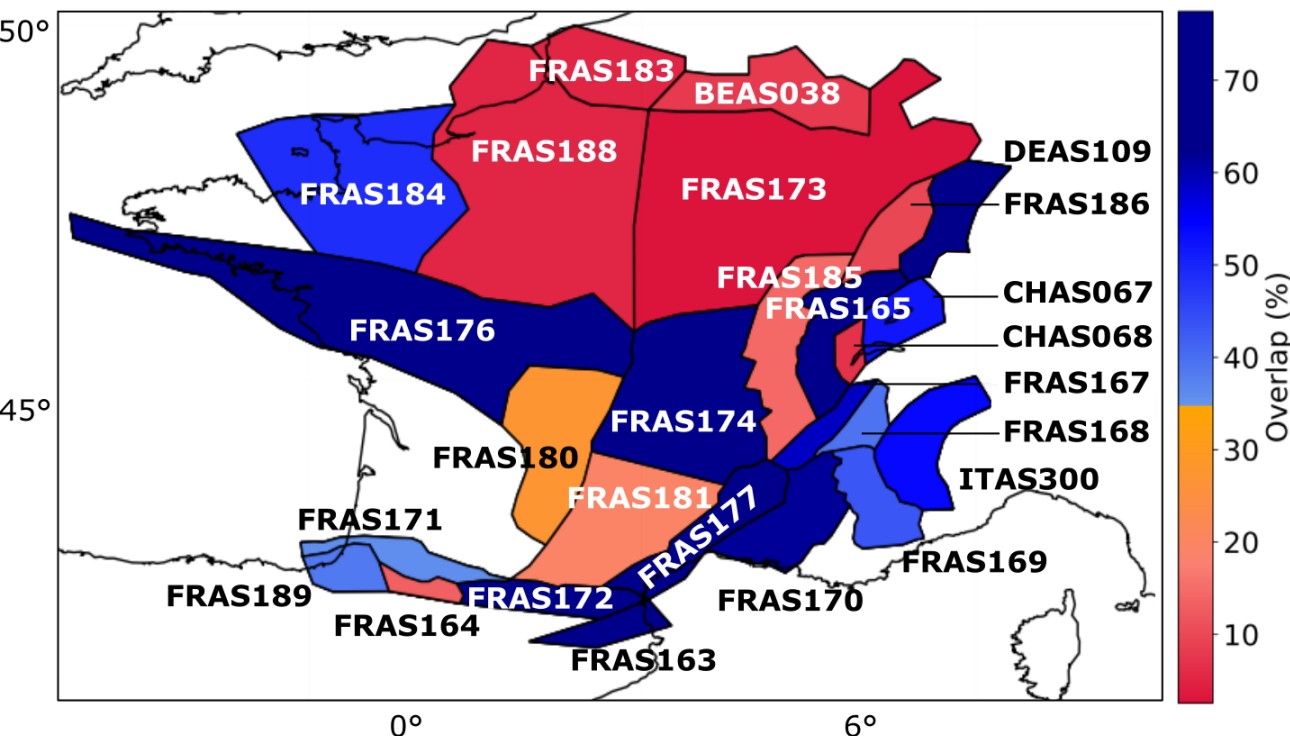

**Figure 15.** Consistency between between $\dot{M}_{0G}$ and $\dot{M}_{0S}$ distributions in France (estimation of the overlap between distributions), at the scale of the area zource zone. Source zone acronyms as in ESHM20 model.



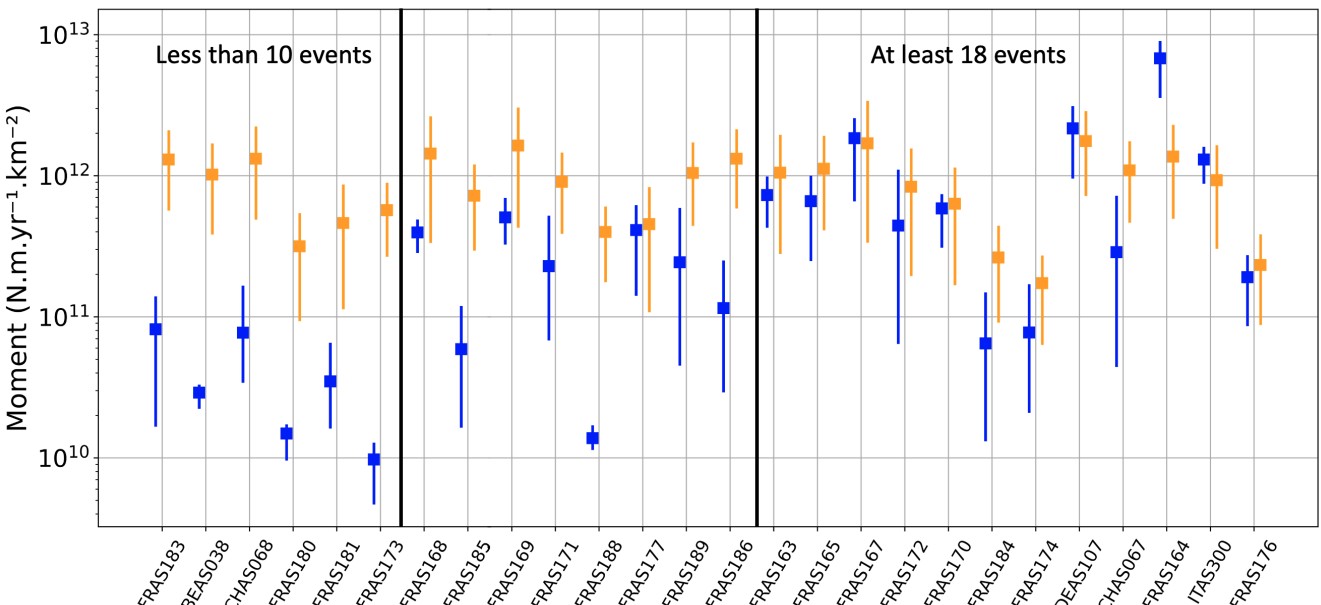

**Figure 16.** Comparison of geodetic moment rate (orange) and seismic moment rate (blue, inferred from the ESHM20 earthquake recurrence model), for source zones in France or on the border. Mean values and percentiles $16^{th}$ and $84^{th}$. The order from left to right correspond to an increasing number of events used for establishing the earthquake recurrence model, less than 10 events for sources FRAS183 to FRAS173, and 30 to 78 events for sources FRAS174 to FRAS176

## 2 Conclusions

We compared the consistency between seismic and geodetic moment rates at the scale of Europe based on strain rate models
(Piña-Valdés et al. (2022)) and long-term earthquake rate forecast from the ESHM20 seismogenic source model (Danciu et al. (2021)). The uncertainties involved in the calculation of the strain rates and associated moment rates are quantified. The obtained distribution for the geodetic moment rate is compared to the distribution inferred from the exploration of the ESHM20 source model logic tree. We observed that in moderate to high seismic activity zones such as the Apennines, Greece, the Balkans and the Betics, the overlap between the distributions of seismic and geodetic moment rates was sufficient to demonstrate a first-
order compatibility. However, local differences between the two distributions were observed when seismicity is concentrated in areas with small characteristic distances (i.e., small areas/faults). This suggests that in some cases, the scale of the source zones is not suitable for comparing seismic moment rate and and geodetic moment rate. As expected, considering larger spatial scale and conducting the same analysis at the macrozone scale yielded improved consistency.

In very low activity zones, such as in Fennoscandia, where the geodetic signal is dominated by glacial isostatic adjustment,
the two estimates (seismic and geodetic) are not compatible, with a geodetic moment much larger than the seismic moment, most probably because in these areas the strain does not represent the long-term tectonic loading, but a transient signal that started at the last glacial maximum, 20,000 years ago. In regions of low-to-moderate seismic activity (e.g., France, Germany,



Spain), the geodetic strain may be representative of the current horizontal tectonic stresses. In some of these areas, the seismic and geodetic moment rate estimates proved to be consistent, taking into account uncertainties, whereas in others the geodetic moment results much larger than the seismic moment. In the area source zones where the ESHM20 earthquake recurrence models are well constrained (with >30 earthquakes used to constrain the models), we observed that the distributions of seismic and geodetic moments tend to overlap. This observation can also be extended to the inclusion of faults to constrain the earthquake recurrence model. Even in area source zones characterized by low strain rates, those where the ESHM20 model includes a sufficient amount of faults and earthquakes used to constrain the earthquake recurrence model, show a good consistency between the geodetic and the seismic moment rates. On the contrary, zones that show no compatibility between the geodetic and the seismic moment rates, are all characterized by a minimum fault density and less than 30 earthquakes used to constrain the earthquake recurrence model. The inclusion of active faults may therefore strengthen the earthquake recurrence model in areas that are characterized by both a slow deformation rate and rare seismic events. This may also suggest that in areas where enough active faults are mapped, the strain rates might be representative of the long term tectonic loading, even if little seismic activity is available to constrain the earthquake recurrence model. More work is needed to understand the consistencies or discrepancies obtained between strain rate based moments and moments relying on the long-term recurrence models built for PSHA.

*Author contributions.* BDJ : data analysis, software, visualisation, writing and editing, AS and CB : data analysis, supervision and research initiation process, writing and editing, JPV : data analysis, processing and review and LD : data sharing and review

*Competing interests.* no competing interests are present

*Acknowledgements.* P. Gueguen, N.D'Agostino, A. Walpersdorf and A. Pothon for insightful discussions This study was funded by the AXA Research Fund supporting the project New Probabilistic Assessment of Seismic Hazard, Losses and Risks in Strong Seismic Prone Regions - SubRisk.





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
