# Peer review of "Consistency between the Strain Rate Model and ESHM20 Earthquake Rate Forecast in Europe: insights for seismic hazard"

_EGUsphere, 2024_

## Referee Comment (RC2)

**Review of the manuscript egusphere-2024-787**

The paper egusphere-2024-787 "Consistency between the Strain Rate Model and ESHM20 Earthquake Rate Forecast in Europe: insights for seismic hazard" presents an approach to compare moment rates computed from geodetic and geological observations also accounting for their uncertainties. Although geodetic observations are not still routinely used to assess the seismic hazard in a region due to the lack of a long-term series of geodetic measurements, models based on geodetic observations have been shown to provide forecasting skills where traditional methods to assess seismic rate models have not (e.g., Rhoades et al., 2017; Rollins and Avouac, 2019; Gerstenberger et al. 2020). In this context, this manuscript is a step forward in this direction.

There are a few adjustments, which could improve the manuscripts.

1) In my opinion, the introduction section should explain better why incorporating geodetic observations is important in the development of seismic rate models and provide examples of where this is applied, including a brief description of the approaches used there.

2) The section conclusion is a simple summary of the results discussed in the previous section. Although this section should emphasize the main result, it should also highlight the strengths and limitations of the study and give future research directions for the full inclusion of geodetic observations in the seismic rate model.

3) How do the geodetic measurements computed in this manuscript compare with the geodetic model for Italy in Meletti et al. (2021)? Are there any other regions in Europe and the Mediterranean area (such as Turkey and Greece), which include a geodetic model in the seismic hazard model? If so, it can be compared with the results of this work.

4) When the authors define the logic tree for the calculations of the geodetic moment rates, some of the alternative models and parameters should be justified better, e.g. the alternative values for the seismogenic thickness and the dip values of 25° and 65°.

5) I find the extensive use of parentheticals often detracts from the readability of the paper. I believe in many cases the parenthetical could be incorporated into the sentence, making it flow better and more readable, or it could be eliminated. Also, the standard way of citing references is:
   - (e.g. Stirling et al., 2012; Field et al., 2014; Beauval et al., 2018)
   - (Woessner et al., 2015)
   - Etc

6) When an acronym is used for the first time in the text, it should be explained, e.g. ESHM20 in the abstract, EFSM80 in line 74, VISR in line 116, GIA in the caption of Figure 6.

7) For the unit, the notation with the dots (e.g. N.m.yr$^{-1}$.km$^{-2}$.) seems to be strange. I would suggest the authors check the notation format for EGUsphere. Also, I would suggest checking the punctuation throughout the manuscript, specifically the use of comma. Below I indicated some of these issues. Finally, when a figure is cited, it should have the capital letter without "the".

Below there are a few (technical or editorial) comments on the manuscript.

Lines 41-42: Provide references for "Indeed, the tectonic loading recorded by geodesy should be proportional to the energy released during earthquakes, under the assumption that the earth's crust behaves elastically".

Line 67: Include ":" after "of two components". Furthermore, the authors should briefly explain what these two components are.

Line 69: Include where the "deep and subduction earthquakes" occur.

Line 78: In "The area source model consists of cross-border harmonized seismogenic sources which geometry is guided" "which" seems to be wrong. Probably it should be replaced with whose.

Lines 79-80: Replace "For every area source," with "For each areal source,".

Line 80: Replace "Gutenberg magnitude-frequency distribution" with "Gutenberg-Richter magnitude-frequency distribution" and "established" with computed or evaluated.

Line 83: "form 2" should have a capital letter, i.e. Form 2.

Line 87: Explain what a corner frequency (Mc) is. Is it the completeness magnitude?

Line 91: Leonard (2015) does not seem the appropriate citation since it is a reply to an article. The authors should use Leonard (2010) and/or its update for stable continental regions Leonard (2014).

Line 92: The smoothed seismicity model and the adaptive kernels should be briefly explained to make this manuscript a stand-alone article.

Line 106: How large is the "spatial cell"? Do the results change as the dimensions of the spatial cells change?

Line 114: Replace "the work done by Piña-Valdés et al. (2022)" with "the work of Piña-Valdés et al. (2022)".

Line 118: Replace "the algorithm uses as inputs the discretized geodetic observations" with "the algorithm uses the discretized geodetic observations as inputs".

Line 138: List the "number of decisions". These then are explained afterwards.

Formula 3: Are $n_{cells}$ and n the same? If so, use the same notation; otherwise, explain them.

Formulas 6-8: Explain all the elements in the notations, for example, A does not seem to be described.

Line 163: Replace "geometric coefficient, it depends" with "geometric coefficient, which depends".

Line 170: How did the authors decide the values of 25° and 65° for the dip. Are these values applied to the sources in France or to the entire European source model?

Line 173: How do the values of 5, 10, and 15 km for the seismogenic thickness be chosen?

Line 180: Replace "obtain a distribution for the moment rate" with "obtain a distribution of the moment rate".

Line 181: Explain how 12 from"the 12 difference preprocessing parameters" comes from.

Lines 190-191: Is there a reason why the authors chose southern Brittany and northern Tuscany?

Lines 199-207: Other parameters show differences between the three selected zones in Figure 5. For example, the class A, AB, and ABC, the spatial weighting, and the Mog

equations. The authors should include these features in the text and explain possible reasons for these differences.

Figure 4: I would suggest including also the weights associated with each branch in Figure 4a and explaining how they were defined.

Lines 212: The earthquake catalogue used for the ESHM20 does not extend over several centuries in the entire region under investigation. In central and north Europe the catalogue is only a few hundred years long, even less in offshore regions. I would suggest rephrasing this sentence.

Figure 5: How is the full distribution (grey lines) computed? Is it a weighted mean of all branches? Include the labels in the y-axis of the top plot. In the caption of this figure, a word is missing after "full exploration of the tree".

Figure 6: Include the name of the zones in this figure.

Line 214: Replace eq with earthquake and add as between "historical seismicity" and "well as on a wider". Is it possible to explain what the "analysis of the seismogenic potential of the area" is?

Lines 215-216: Include references for "The earthquake rate forecast model also includes our current knowledge about active faults".

Lines 222-223: The results for Central Apennines, Greece, and Turkey do not support this sentence. For those areas, the seismic moment rates seem to be larger than the geodetic moment rates.

Figure 7: Is the mean seismic moment in Figure 7a computed from the Gutenberg-Richter frequency-magnitude distribution for the entire ESHM20 source model or does it account for only the areal source model or the smoothed model and the fault model? This should be clearly indicated in the text.

Line 256: Remove the comma in "We quantify the overlap between the geodetic and the seismic distributions, for all area sources".

Line 257: Replace "the overlap between the distributions is usually increasing with closer mean moment values" with "the overlap between the distributions usually increases with closer mean moment values".

Line 259: Provide examples for "elsewhere the fit is quite poor".

Figure 10: Which are the zones associated with the reddish dots 5 and 8?

Line 264: Deleted the comma in "the smoothed seismicity model, for the underlying".

Line 269: What are the reasons for the lack of good fit in Spain when the macrozones are used? Which are the specific criteria used to assess that the overall fit is good from Figure 11? I would say that the fit is relatively good only for the highly seismic regions, not for central and northern Europe looking at Figure 11.

Section 1.3.3: Out of curiosity, in which category do the zones for the UK fall?

Line 292: Remove "the" before "Figure 12".

Line 343: Replace "ESHM20) : We observe" with "ESHM20. We observe"

Lines 335-353: In this paragraph, I would suggest including the examples of zones to strengthen the argument here. For example, "in the area zones that include faults than in area zones that do not include any fault in the model" [which zones? Where they are?] and "in zones with lower strain" [which zones? Where they are?].

Line 351: Remove "by the geologists" because it is obvious.

Figure 12: Why is the caption in bold?

Figure 13: The rhomboidal symbols to indicate the zone affected by GIA are not clear from the figure. I would suggest to change shape and/or colour.

Line 361: Where are a and b in Figure 8? And the profile AB in Figure 8?

Line 373: Is the citation of Figure 14b correct?

Lined 378- 380: Include references for "In Italy the density of GPS stations is quite high with an interstation distance of 20km on average, and the network should capture any spatial details larger than 30km in the deformation field. The observed difference in spatial distribution between the".

Line 382: Include references for the elastic rebound theory.

Lines 383-385: Include references for "During the interseismic period, the deformation associated with the loading is usually modeled as a fault that is locked down to a given depth and that is creeping at the loading rate at greater depths.".

Line 393: There is a word missing in "can be meaningful only if led at a large enough spatial scale.", probably "they" before "led".

Figure 14: There are too many brackets in the caption of this figure and it is difficult to understand what the plots show.

Line 407: Replace "obtained" with estimated or computed.

References

Gerstenberger, MC, et al. 2020. Probabilistic seismic hazard analysis at regional and national scales: state of the art and future challenges, Review of Geophysics, DOI:10.1029/2019RG000653.

Leonard, M. 2010. Earthquake fault scaling: Relating rupture length, width, average displacement, and moment release, Bulletin of the Seismological Society of America, Vol. 100, no. 5, 1971–1988.

Leonard, M. 2014. Self-Consistent Earthquake Fault-Scaling Relations: Update and Extension to Stable Continental Strike-Slip Faults, Bulletin of the Seismological Society of America, Vol. 104, No. 6, pp. 2953–2965.

Meletti, C, et al. 2021. The new Italian Seismic Hazard Model (MPS19), Annals of Geophysics,  64(1), SE112, 2021, doi:10.4401/ag-8579.

Rhoades, DA, Christophersen, A, and Gerstenberger, MC. 2017. Multiplicative earthquake likelihood models incorporating strain rates. Geophysical Journal International, 208(3), 1764–1774.

Rollins, C, and Avouac, J-P. 2019. A geodesy- and seismicity-based local earthquake likelihood model for central Los Angeles. Geophysical Research Letters, 46, 3153–3162.

---

## Author Comment (AC1)

Dear Editor, Associate Editor, and Reviewers,

We would like to express our gratitude for the valuable and constructive comments provided. We have addressed most of the critical points raised and incorporated many suggestions to improve the readability of the article.

Responses to reviewers comments can be found below. The questions are presented in black, and our answers are shown in blue. Each modification to the manuscript text is indicated in green.

Thank you for considering this revised version of our manuscript.

Sincerely,
Bénédicte Donniol Jouve on behalf of the co-authors

**Reviewer #1:**

This is a quite interesting paper which focuses on the comparison of geodetic and seismic moment rates across Europe. The approach is successful in spite of the large area examined and its high seismotectonic heterogeneity.

A major issue which needs revision is the organization of the paper. In lines 61-64 the authors claim that "In a first step, we present the datasets and methods used to compute the seismic and geodetic moments integrated in space and time and to explore the uncertainties. Next, we compare the estimated seismic and geodetic moments in the different seismogenic source zones of ESHM20 that covers the Euro-Mediteranean region. We then discuss the parameters that influence the most the compatibility in both high and low-to moderate seismic activity." However, the overall structure of the paper is inconsistent with the claim. Namely, there is a main section "1. Introduction" and all the material of the paper is presented within this section with sub-sections numbers ranging from 1.1 to 1.5. Section 1 is followed by section "2. Conclusions".

I recommend the drastic reorganization of the paper's structure in a way that makes clear the "real" Introduction, which should be followed by an appropriate number of sections, possibly four, devoted to "Methods and Data", "Results", "Discussion", and "Conclusions".

We agree with your comment, there is indeed an issue there. Thank you very much for pointing it out. We have added section titles at line 70 ('Methods and Data'), line 226 ('Results'), and line 378 ('Discussion'), in accordance with your suggestions.

Other comments.

47-49. "In the Hellenic arc, Jenny et al. (2004), found that the maximum magnitudes required for the earthquake recurrence models to be moment-balanced were unrealistic and concluded that a large part of the strain is released in aseismic processes". However, this fundamental result has been supported by previous authors, including Papadopoulos (1989) and Becker & Meier (2010).

Thank you for your comment. We have included the citation to Papadopoulos (1989) in the text.

227-228. "If earthquake catalogs of much longer time windows were available (e.g. 100,000 years), would the spatial distribution of the seismic moment rates be more alike the spatial distribution of the geodetic moment rates?" This critical question is not replied. Do the authors have a reply to that?

We think this question is fundamental and should be posed, but we do not have an answer (unfortunately nobody has the answer).

305. The last glacial maximum should be ~20,000 years.

Thank you for pointing out this punctuation error. We have made the requested correction.

In Figure 2, the black polygons representing area sources in most cases are not recognizable. Is it possible to improve its? Similarly, Figure 7 and subsequent figures need improvement.

We acknowledge that to identify all details, in some cases it is necessary to zoom in. The purpose of this figure is to provide general information at the European level.

A more readable plot of the source zones is presented in Figure 6. To aid readers in better identifying these zones, we completed the captions of Figures 2, 7, and 8 with the following sentence : 'Area source zone polygons are also displayed in Figure 6.'

References

Becker, D. Meier, Th., 2010. Seismic Slip Deficit in the Southwestern Forearc of the Hellenic Subduction Zone. Bulletin of the Seismological Society of America, 100 (1): 325–342. doi: https://doi.org/10.1785/0120090156

Papadopoulos, G.A., 1989. Seismic and volcanic activities and aseismic movements as plate motion components in the Aegean area, Tectonophysics, Volume 167, Issue 1, Pages 31-39, https://doi.org/10.1016/0040-1951(89)90292-8.

**Reviewer #2:**
The paper egusphere-2024-787 "Consistency between the Strain Rate Model and ESHM20 Earthquake Rate Forecast in Europe: insights for seismic hazard" presents an approach to compare moment rates computed from geodetic and geological observations also accounting for their uncertainties. Although geodetic observations are not still routinely used to assess the seismic hazard in a region due to the lack of a long-term series of geodetic measurements, models based on geodetic observations have been shown to provide forecasting skills where traditional methods to assess seismic rate models have not (e.g., Rhoades et al., 2017; Rollins and Avouac, 2019; Gerstenberger et al. 2020). In this context, this manuscript is a step forward in this direction.
There are a few adjustments, which could improve the manuscripts.

We are very grateful to reviewer #2 for his/her very thorough review and careful reading that helps substantially to improve the manuscript.

1. 1) In my opinion, the introduction section should explain better why incorporating geodetic observations is important in the development of seismic rate models and provide examples of where this is applied, including a brief description of the approaches used there.

Thank you very much for the suggestion. We have added the following sentence at line 52:
"These moment-balanced earthquake recurrence models can be combined to ground-motion models to quantify probabilistic seismic hazard (e.g. Stevens and Avouac 2021). To our knowledge, in Europe, the only seismic hazard model that integrates a source model based on strain rates is the new Italian hazard model (Meletti et al. 2021). Gridded seismicity rates are inferred from geodetic strain rates, following in particular the method of Carafa et al. (2017). A number of studies have demonstrated how geodetic strain rates correlate with seismic rates (e.g. Zeng et al. 2018)."

2) The section conclusion is a simple summary of the results discussed in the previous section. Although this section should emphasize the main result, it should also highlight the strengths and limitations of the study and give future research directions for the full inclusion of geodetic observations in the seismic rate model.

We have revised the last paragraph (lines 454-458):
"More work is needed to understand the consistencies or discrepancies obtained between strain rate based moments and moments relying on the long-term recurrence models built for PSHA. Some parameters such as the effective seismic thickness will need to be better evaluated to improve the estimation of the moment rate from strain rates. Nonetheless, our work demonstrates the strong correlation between long-term seismic moment rates and geodetic moment rates, paving the way for the wider integration of geodetic data in probabilistic seismic hazard model."

3) How do the geodetic measurements computed in this manuscript compare with the geodetic model for Italy in Meletti et al. (2021)? Are there any other regions in Europe and the Mediterranean area (such as Turkey and Greece), which include a geodetic model in the seismic hazard model? If so, it can be compared with the results of this work.

Thank you for the suggestion. While it would indeed be beneficial to compare with the cited geodetic model, such a comparison is beyond the scope of our current article. Nonetheless we have added the following sentence to the text: "To our knowledge, in Europe, the only seismic hazard model that integrates a source model based on strain rates is the new Italian hazard model (Meletti et al. 2021)." (l53-54)

4)  When the authors define the logic tree for the calculations of the geodetic moment rates, some of the alternative models and parameters should be justified better, e.g. the alternative values for the seismogenic thickness and the dip values of 25° and 65°.

Thank you for the question. The objective of using these two coefficients was to account for their variability across Europe, given that our datasets were not precise enough to estimate them individually for each source zone. Therefore, we chose these values as minimum and maximum bounds within which these coefficients could reasonably vary.

Regarding the geometric coefficient Cg, we selected two values for fault dip angles, 25° and 65°, representing thrust and normal faults, respectively.

Regarding the elastic thickness H, we aim at exploring the uncertainty related to this parameter, we explore the range 5 to 15 km following previous efforts to convert strain rates into earthquake rates in different regions of the world (Ward 1998, Pancha et al. 2006, D'Agostino 2014, Carafa et al. 2017; Stevens and Avouac 2021).

We have therefore added the following text, for more clarity:

L180 : Here we consider two values, 2 and 2.6, which is the range corresponding to a dip between 25° and 65°, representing standard thrust and normal faults, respectively.

L183 : 'Whereas for the seismogenic thickness (H in Equations 6 to 8), we consider here the elastic thickness, i.e. the average thickness over which a region's principal faults store and release seismic energy (Ward, 1998). Only a fraction of the frictional slip takes place during earthquakes (Bird et al. 2002).  Mazzotti et al. (2005) define the "effective seismic thickness" as the thickness of the crust where deformation is fully accommodated by seismicity.  In an application in eastern North America, they show that this effective seismic thickness may represent only 40% of the seismogenic thickness based on maximum and minimum depths of earthquakes.   The thickness considered in the literature to evaluate seismic moment release from strain rates usually varies between 10 and 15km. Pancha et al. (2006) used a fixed seismogenic thickness of 15km throughout the Basin and Range region in Western US. D'Agostino et al. (2014) applied a thickness of 10± 2.5km throughout the Apennines in Italy, whereas Stevens and Avouac (2021) considered 15km in the India-Asia collision zone. Carafa et al. (2017) estimated average coupled thicknesses between ~3 and ~8km for faults in Italy.  As there is considerable uncertainty, we use three alternative values (5, 10, and 15 km) and propagate this uncertainty up to the geodetic moment rate estimates.'

5)  I find the extensive use of parentheticals often detracts from the readability of the paper. I believe in many cases the parenthetical could be incorporated into the sentence, making it flow better and more readable, or it could be eliminated. Also, the standard way of citing references is:
    o   - (e.g. Stirling et al., 2012; Field et al., 2014; Beauval et al., 2018)
    o   - (Woessner et al., 2015) - Etc

We have replaced all citations with the standard references you mentioned. Additionally, we have removed unnecessary parentheses.

L 24-26 : 'However, fault databases are known to be incomplete, even in the best characterized regions, and earthquakes may occur on unknown faults, as demonstrated by several earthquakes in the past, as the two 2002 Mw 5.7 Molise earthquakes (Valensise et al., 2004) in Italy or the Darfield Mw7.1 earthquake in New Zealand (Hornblow et al., 2014)'

L34-37 : Along major interplate faults, such as subduction zones or lithospheric strike slip faults, interseismic velocities measured by GNSS are now commonly used to constrain the slip deficit on the fault associated with locking in between large seismic events, also referred to as interseismic coupling.

6) When an acronym is used for the first time in the text, it should be explained, e.g. ESHM20 in the abstract, EFSM80 in line 74, VISR in line 116, GIA in the caption of Figure 6.

Done. The meaning of the acronyms have been added in the manuscript.

7) For the unit, the notation with the dots (e.g. N.m.yr⁻¹.km⁻². ) seems to be strange. I would suggest the authors check the notation format for EGUsphere. Also, I would suggest checking the punctuation throughout the manuscript, specifically the use of comma. Below I indicated some of these issues. Finally, when a figure is cited, it should have the capital letter without "the".

We have updated the notation from N.m.yr⁻¹.km⁻² to N m yr⁻¹ km⁻² in the manuscript and in the figures. Additionally, we have revised the figure citations as advised.

Below there are a few (technical or editorial) comments on the manuscript.

Lines 41-42: Provide references for "Indeed, the tectonic loading recorded by geodesy should be proportional to the energy released during earthquakes, under the assumption that the earth's crust behaves elastically".

Done, we have added the reference (Reid, 1910).

Line 67: Include ":" after "of two components". Furthermore, the authors should briefly explain what these two components are.

We have corrected the text accordingly. 'The regional hazard model consists of two main components : a seismogenic source model that forecasts earthquakes in space, time and magnitude, and a ground-motion model that predicts the ground-motions that these earthquakes may generate.'

Line 69: Include where the "deep and subduction earthquakes" occur.

We have corrected the text accordingly. 'The earthquake rate forecast includes all earthquake types, i.e., crustal, deep (Vrancea region, Romania), and subduction (Hellenic, Cyprian, Calabrian and Gibraltar Arcs) earthquakes. In this paper we focus on the contribution of crustal shallow seismogenic sources that can be straightforward compared to surface strain rate.'

Line 78: In "The area source model consists of cross-border harmonized seismogenic sources which geometry is guided" "which" seems to be wrong. Probably it should be replaced with whose.

We have modified the text accordingly.

Lines 79-80: Replace "For every area source," with "For each areal source,".

We have modified the text accordingly.

Line 80: Replace "Gutenberg magnitude-frequency distribution" with "Gutenberg-Richter magnitude-frequency distribution" and "established" with computed or evaluated.

We have modified the text accordingly.

Line 83: "form 2" should have a capital letter, i.e. Form 2.

We have modified the text accordingly.

Line 87: Explain what a corner frequency (Mc) is. Is it the completeness magnitude?

We now provide an explanation. The corner magnitude Mc is defined as the magnitude at which there is a bending in the earthquake recurrence model. In the ESHM20 model, this parameter has been estimated based on the observed maximum magnitude (see technical report by Danciu, 2021, pp. 48-49 for more details)

Line 91: Leonard (2015) does not seem the appropriate citation since it is a reply to an article. The authors should use Leonard (2010) and/or its update for stable continental regions Leonard (2014).

That was indeed an error. We have modified the reference accordingly.

Line 92: The smoothed seismicity model and the adaptive kernels should be briefly explained to make this manuscript a stand-alone article.

Thank you for your comment. We have revised the text to: 'The smoothed seismicity model is built from the earthquake catalog, it forecasts earthquake rates within spatial cells'.

Line 106: How large is the "spatial cell"? Do the results change as the dimensions of the spatial cells change?

The cell width is 0.1°. We cannot test the impact of the cell width considered as we did not define them; they originate from the ESHM20 model.

Line 114: Replace "the work done by Piña-Valdés et al. (2022)" with "the work of Piña- Valdés et al. (2022)".

We have modified the text accordingly.

Line 118: Replace "the algorithm uses as inputs the discretized geodetic observations" with "the algorithm uses the discretized geodetic observations as inputs".

We have modified the text accordingly.

Line 138: List the "number of decisions". These then are explained afterwards.

Thank you. The sentence has been revised in accordance with your suggestion : 'While applying the algorithm VISR, a number of decisions are required that may impact horizontal strain rates estimates : the distance and spatial weighting scheme, and the weighting threshold implied in the spatial inversion.'

Formula 3: Are ncells and n the same? If so, use the same notation; otherwise, explain them.

They are not the same. One refers to the smoothed seismicity model (ESHM20), the other to the strain rate map (Piña-Valdes et al.). There is no need for those numbers to be equal.
We have clarified the text:
l160 'With ncells the number of cells considered.'

Formulas 6-8: Explain all the elements in the notations, for example, A does not seem to be described.

A is now defined.

Line 163: Replace "geometric coefficient, it depends" with "geometric coefficient, which depends".

Thank you for your careful reading; we have modified the text accordingly.

Line 170: How did the authors decide the values of 25° and 65° for the dip. Are these values applied to the sources in France or to the entire European source model? Line 173: How do the values of 5, 10, and 15 km for the seismogenic thickness be chosen?

These values are defined at the scale of Europe. We provide a more precise answer to this subject in question 4.

Line 180: Replace "obtain a distribution for the moment rate" with "obtain a distribution of the moment rate".

Thank you, we have modified the text accordingly.

Line 181: Explain how 12 from "the 12 difference preprocessing parameters" comes from.

We examine three distinct categories of stations (A, AB, and ABC). For each category, we evaluate four different outlier radii (50, 100, 150, and 200). Consequently, this results in a total of 12 (3x4) distinct preprocessing parameter sets being considered.

Lines 190-191: Is there a reason why the authors chose southern Brittany and northern Tuscany?

Models in these two area source zones are rather well constrained.

Lines 199-207: Other parameters show differences between the three selected zones in Figure 5. For example, the class A, AB, and ABC, the spatial weighting, and the Mog equations. The authors should include these features in the text and explain possible reasons for these differences.

Thank you for your comment. We have revised the text to: 'The results show that the uncertainty on the effective seismic thickness controls the overall moment rate variability, for all area source zones. The geodetic moment rate exhibits a linear variation with both the effective seismic thickness and the shear modulus. Except for the shear modulus for which a limited range of values is explored, all other parameters uncertainties also contribute to the overall variability. It is interesting to note that the exact selection of GNSS stations, controled by the selection steps related to the Class and the Radius Outlier, has an influence on the moment rate estimates only in low seismicity areas (Fennoscandia and Southern Brittany), but no impact in the moderate to high seismicity regions (such as northern Tuscany). This phenomenon can be attributed to the high strains in high-deformation zones, where even lower-quality stations provide accurate measurements at a first-order approximation. Conversely, in low-deformation areas, the measured signal is close to the noise level (hence, highly uncertain). Consequently, the exclusion or inclusion of one or more stations has a substantial impact.

Furthermore, it is noteworthy that the parameters involved in the spatial inversion, particularly the distance weighting scheme, have a significant impact on the overall uncertainty. This impact is more pronounced in regions with a relatively high density of GNSS stations, such as northern Tuscany and southern Brittany. The Gaussian function reduces data weight with distance faster than the Quadratic function, which can yield a smoother solution when dealing with heterogeneous data. In regions with a high density of stations, this may lead to higher strain rates calculated using the Gaussian function than those obtained with the Quadratic function. Additionally, the weighting threshold, which controls the smoothing of the solution, naturally has a greater impact in regions with a higher station density. Another parameter with a non-negligible impact on the total uncertainty is the equation used to calculate the geodetic moment.'

Figure 4: I would suggest including also the weights associated with each branch in Figure 4a and explaining how they were defined.

We are using equal weights for the exploration of uncertainties.

Lines 212: The earthquake catalogue used for the ESHM20 does not extend over several centuries in the entire region under investigation. In central and north Europe the catalogue is only a few hundred years long, even less in offshore regions. I would suggest rephrasing this sentence.

Thank you for your comment. We have revised the text to:
'The ESHM20 earthquake rate forecast relies on earthquake catalogs extending over several centuries in most regions of the study area'

Figure 5: How is the full distribution (grey lines) computed? Is it a weighted mean of all branches? Include the labels in the y-axis of the top plot. In the caption of this figure, a word is missing after "full exploration of the tree".

The grey line represents the overall distribution of all the solutions explored in Figure 4a.

Figure 6: Include the name of the zones in this figure.

We have included the name of the zones in Figure 6.

Line 214: Replace eq with earthquake and add as between "historical seismicity" and "well as on a wider". Is it possible to explain what the "analysis of the seismogenic potential of the area" is?

We have modified as suggested.
A wider analysis of the seismogenic potential of the area refers to accounting for geology, geodynamics, geomorphological, and seismological datasets.

Lines 215-216: Include references for "The earthquake rate forecast model also includes our current knowledge about active faults".

The reference to Basili et al. NHESS has been added.

Lines 222-223: The results for Central Apennines, Greece, and Turkey do not support this sentence. For those areas, the seismic moment rates seem to be larger than the geodetic moment rates.

Do you refer to the sentence: "Overall, geodetic moment rates appear larger or equal to seismic moment rates, similarly to the findings of many previous studies"? This is the general trend, but they are exceptions.

Figure 7: Is the mean seismic moment in Figure 7a computed from the Gutenberg-Richter frequency-magnitude distribution for the entire ESHM20 source model or does it account for only the areal source model or the smoothed model and the fault model? This should be clearly indicated in the text.

In the Figure 7's caption, it is explicitly indicated that Figure 7a shows the ''Mean geodetic moment (Mog) based on the strain rates" which is not related to the Gutenberg-Richter frequency-magnitude distribution, but the geodetic observations. The caption also says that Figure 7b corresponds to the "Mean seismic moment estimated from the ESHM20source model logic tree." ESHM20 source model logic tree is the entire logic tree, all branches, area source model, and smoothed seismicity combined with faults.

Line 256: Remove the comma in "We quantify the overlap between the geodetic and the seismic distributions, for all area sources".

We have removed the comma.

Line 257: Replace "the overlap between the distributions is usually increasing with closer mean moment values" with "the overlap between the distributions usually increases with closer mean moment values".

Thank you for your suggestion; we have modified the text accordingly.

Line 259: Provide examples for "elsewhere the fit is quite poor".

We have modified the text as follows.
"In the most seismically active regions in Europe, i.e. in Greece, Italy, the Balkans, as well as in some parts of France and Switzerland, the seismic and geodetic moment estimates are rather consistent (overlap between 35 and 80%, in blue); whereas in most of northern Europe, the fit is quite poor (overlap lower than 30%, in red)."

Figure 10: Which are the zones associated with the reddish dots 5 and 8?

zone 5:
Caption of Figure 10 states : "5: FRAS164," whereas the text states : "The source zone FRAS164 in Western Pyrenees is a small area with high seismic activity in comparison with the neighboring area zones."
zone 8:
Caption of Figure 10 states : "8: CHAS071", we have completed the sentence in the text :
"The source zones CHAS071 (Switzerland), DEAS113 and DEAS109 (Germany) are not as active, but they are very small size area sources."

Line 264: Deleted the comma in "the smoothed seismicity model, for the underlying".

The comma has been deleted.

Line 269: What are the reasons for the lack of good fit in Spain when the macrozones are used? Which are the specific criteria used to assess that the overall fit is good from Figure 11? I would say that the fit is relatively good only for the highly seismic regions, not for central and northern Europe looking at Figure 11.

A more detailed analysis focused on Spain and the local datasets available there would be required to understand why the fit is not good at the macrozone scale. One reason could be that there is insufficient GNSS data for the signal to accurately represent the tectonic loading of the zone.
We have fixed arbitrarily to 35% the overlap threshold for areas in orange-red (poor to very poor fit) with respect to areas in blue (good to very good fit).
Note that the fit results also good in some low-to-moderate seismicity regions (e.g. northwestern France).

Section 1.3.3: Out of curiosity, in which category do the zones for the UK fall?

The UK zones are not included in this section as they are characterized by a geodetic moment significantly higher than the seismic moment.

Line 292: Remove "the" before "Figure 12".

We have modified the text accordingly.

Line 343: Replace "ESHM20) : We observe" with "ESHM20. We observe"

We have modified the text accordingly.

Lines 335-353: In this paragraph, I would suggest including the examples of zones to strengthen the argument here. For example, "in the area zones that include faults than in area zones that do not include any fault in the model" [which zones? Where they are?] and "in zones with lower strain" [which zones? Where they are?].

Figure 1 shows which area source zones have faults included in the model. We have modified the text as follows:
"We group the area zones that include faults on one side, and the area zones that do not include any fault on the other side (see Figure 1)."

Line 351: Remove "by the geologists" because it is obvious.

Text modified, thank you.

Figure 12: Why is the caption in bold?

It was a mistake. We have removed the bold formatting from the caption of this figure.

Figure 13: The rhomboidal symbols to indicate the zone affected by GIA are not clear from the figure. I would suggest to change shape and/or colour.

We modified the figure and changed the symbol to enhance clarity.

Line 361: Where are a and b in Figure 8? And the profile AB in Figure 8?

Thank you, it was a mistake. We refer to the correct Figure now (Figure 14).

Line 373: Is the citation of Figure 14b correct?

Thank you, it was a mistake. We refer to the correct Figure now (14d).

Lined 378- 380: Include references for "In Italy the density of GPS stations is quite high with an interstation distance of 20km on average, and the network should capture any spatial details larger than 30km in the deformation field. The observed difference in spatial distribution between the".

We added the reference Piña-Valdès et al. (2022).

Line 382: Include references for the elastic rebound theory.

We have added the reference (Reid, 1910).

Lines 383-385: Include references for "During the interseismic period, the deformation associated with the loading is usually modeled as a fault that is locked down to a given depth and that is creeping at the loading rate at greater depths.".

Thank you, we have added the reference (Avouac, 2015).

Line 393: There is a word missing in "can be meaningful only if led at a large enough spatial scale.", probably "they" before "led".

Thank you, we have revised the sentence as follows: 'Therefore, in places where source zones enclose fault zones or areas with high seismic activity, the comparison between the geodetic moment and the seismic moment can be meaningful only if it is led at a large enough spatial scale.'

Figure 14: There are too many brackets in the caption of this figure and it is difficult to understand what the plots show.

Thank you, we have revised the caption as follows: 'Spatial variability of geodetic deformation and seismic release in the central Apennines. a) Zoom of Figure 8.a : Mean ratio between Mos and Mog  for area source zones in Central Italy. b) Mean geodetic moment rate per surface unit inferred from strain rates; gray dots : earthquakes in the ESHM20 unified earthquake catalog; blue lines : active faults included in the EFSM20. c and d) Geodetic ($M_{0G}$) and seismic ($M_{0S}$) moment rates per kilometer along the cross-section AB : averaged within the source zones (c), or averaged within bins of 14 km along a 190 km-wide swath profile, represented by the thin gray rectangle (d) Blue arrows : location of the two main faults.'

Line 407: Replace "obtained" with estimated or computed.

We have modified the text accordingly.
Thank you very much for your careful reading.

References
Gerstenberger, MC, et al. 2020. Probabilistic seismic hazard analysis at regional and national scales: state of the art and future challenges, Review of Geophysics, DOI:10.1029/2019RG000653.
Leonard, M. 2010. Earthquake fault scaling: Relating rupture length, width, average displacement, and moment release, Bulletin of the Seismological Society of America, Vol. 100, no. 5, 1971–1988.
Leonard, M. 2014. Self-Consistent Earthquake Fault-Scaling Relations: Update and Extension to Stable Continental Strike-Slip Faults, Bulletin of the Seismological Society of America, Vol. 104, No. 6, pp. 2953–2965.
Meletti, C, et al. 2021. The new Italian Seismic Hazard Model (MPS19), Annals of Geophysics, 64(1), SE112, 2021, doi:10.4401/ag-8579.

Rhoades, DA, Christophersen, A, and Gerstenberger, MC. 2017. Multiplicative earthquake likelihood models incorporating strain rates. Geophysical Journal International, 208(3), 1764–1774.

Rollins, C, and Avouac, J‑P. 2019. A geodesy‑ and seismicity‑based local earthquake likelihood model for central Los Angeles. Geophysical Research Letters, 46, 3153–3162.

**Reviewer #3:**

The study is well-organized and clear. To complete it I suggest inserting some quotes in the introductory part:

ART1 : Nakamura, M., Kinjo, A. Activated seismicity by strain rate change in the Yaeyama region, south Ryukyu. Earth Planets Space 70, 154 (2018).

ART2 : Pappachen, J. et al (2021). Crustal velocity and interseismic strain-rate in the Garhwal–Kumaun Himalaya. Scientific reports, 11(1), 1-13.

ART3 : Zeng, Y. et al (2018). Earthquake potential in California-Nevada implied by correlation of strain rate and seismicity. Geophysical Research Letters, 45.

In the paragraph 1.4 Focus in Italy, please produce a comparative and critical analysis with the following previous studies, focussing on the difference of the applied methods and the conclusion:

ART4 : Riguzzi, et al (2012). Geodetic strain rate and earthquake size: new clues for seismic hazard studies. Physics of the Earth and Planetary Interiors, 206.

ART5 : Farolfi, G., et al (2020). Spatial forecasting of seismicity provided from earth observation by space satellite technology. Scientific reports, 10(1), 1-7.

ART6 : Piombino, A. et al  (2021). Assessing current seismic hazards in Irpinia forty years after the 1980 earthquake: Merging historical seismicity and satellite data about recent ground movements. Geosciences, 11(4), 168.

Once these minor changes have been made, the article can be published.

Thank you for these suggestions.
We have checked the articles listed.
We have added the reference Zeng et al. (2018) in the introduction, as it falls well within the topic addressed in our manuscript.

**Reviewer #4**

The paper deals with the contribution of geodetic monitoring to the probabilistic hazard assessment, by enhancement of the source model. The subject is fascinating and worth to be published. There are, however, certain points in the manuscript that need additional work and corrections. Specific comments are reported, which I hope will contribute to improving its revised version.

MAJOR COMMENTS

1. Line 173: Is it only the thickness or exactly the boundaries (upper and lower depth) of the seismogenic layer? This statement demands more elaboration.
2. Line 175: There are plenty of publications related to your study area where you may take this information. Highly accurate relocated data provide a consistent definition of the seismogenic layer. It seems that you have not taken into account these outcomes from Greece.
3. Line 199: This complies with my comment to take as much precisely as possible the seismogenic layer. These data exist in numerous publications and my suggestion is to take them into account in your calculations.

Thank you very much for these comments. There are different definitions of the seimogenic depths. As underlined by Mazzotti and Adams (2005), the seismic thickness to use in Equations 6 to 8 is a debated parameter with strong epistemic uncertainty. Here we refer to the part of the crust where deformation is fully accommodated by seismicity. There is considerable uncertainty on this parameter, and therefore we have decided to explore the uncertainty by considering 3 alternative values (5, 10, 15km). We have modified the paragraph on the seismogenic depth, as follows:

"Whereas for the seismogenic thickness (H in Equations 6 to 8), we consider here the elastic thickness, i.e. the average thickness over which a region's principal faults store and release seismic energy (Ward, 1998). Only a fraction of the frictional slip takes place during earthquakes (Bird et al. 2002).  Mazzotti et al. (2005) define the "effective seismic thickness" as the thickness of the crust where deformation is fully accommodated by seismicity.  In an application in eastern North America, they show that this effective seismic thickness may represent only 40% of the seismogenic thickness based on maximum and minimum depths of earthquakes. The thickness considered in the literature to evaluate seismic moment release from strain rates usually varies between 10 and 15km. Pancha et al. (2006) used a fixed seismogenic thickness of 15km throughout the Basin and Range region in Western US. D'Agostino et al. (2014) applied a thickness of 10±2.5km throughout the Apennines in Italy, whereas Stevens and Avouac (2021) considered 15km in the India-Asia collision zone. Carafa et al. (2017) estimated average coupled thicknesses between ~3 and ~8km for faults in Italy.  As there is considerable uncertainty, we use three alternative values (5, 10, and 15 km) and propagate this uncertainty up to the geodetic moment rate estimates."

Bird et al. 2002 : Bird, P., Kagan, Y.Y. & Jackson, D.D., 2002. Plate tectonics and earthquake potential of spreading ridges and oceanic transform faults, in Plate Boundary Zones, Geodynamics Series 30, pp. 203–218, eds Stein, S. & Freymueller, J.T., American Geophysical Union, Washington, DC.

Therefore, we have added the following text for clarity:

L 170 : Here we consider two values, 2 and 2.6, which is the range corresponding to a dip between 25° and 65°, representing standard thrust and normal faults, respectively.

Last paragraph of the conclusions : 'More work is needed to understand the consistencies or discrepancies obtained between strain rate based moments and moments relying on the long-term recurrence models built for PSHA. Some parameters such as the effective seismic thickness will need to be better evaluated to improve the estimation of the moment rate from strain rates. Nonetheless, our work demonstrates the strong correlation between long-term seismic moment rates and geodetic moment rates, paving the way for the wider integration of geodetic data in probabilistic seismic hazard model.'

4. Please, provide the outcomes from the fast deforming areas, alike Greece and western Turkey.

We are sorry, but we do not understand what part of the manuscript you are referring to, nor what outcomes you may have in mind. We cannot answer this comment.

5. Line 213: How are you evaluating the largest possible earthquake in each source? This is a very delicate issue and must be considered with caution. Even in areas with a wealth of historical data, like in Aegean, the definition of Mmax demands much elaboration.

Actually, we didn't evaluate the maximum magnitude in each source. As explained in the manuscript, we use the ESHM20 earthquake recurrence model, the full source model logic tree. Information on the estimation of the maximum magnitude of source zones can be found in Danciu et al. 2024 (https://doi.org/10.5194/egusphere-2023-3062) and Basili et al. 2024 (https://doi.org/10.5194/nhess-2023-118).

6. Line 216: Fie how many faults have you got documented traces? They are very rare for earthquakes of m~6.0 or smaller. You must support this input for each fault segment.

The number of faults varies with the zone considered. We did not build the model ourselves but directly used the one proposed by ESHM20. For more information, please refer to the following:

Basili R., et al . https://doi.org/10.13127/efsm20

Basili, R., et al : The European Fault-Source Model 2020 (EFSM20): geologic input data for the European Seismic Hazard Model 2020, Nat. Hazards Earth Syst. Sci. Discuss. [preprint], https://doi.org/10.5194/nhess-2023-118, in review, 2023

Danciu et al. (2024, https://doi.org/10.5194/egusphere-2023-3062),

Danciu et al. 2021 EFEHR technical report (https://doi.org/10.12686/a15).

7. Line 216: How do you know the extension in depth? This is based on highly accurate relocated seismicity, but this component is missing in your work.

Similarly, we used the work done by ESHM20. Please refer to their report for more information:

Basili R., et al . https://doi.org/10.13127/efsm20

Basili, R., et al : The European Fault-Source Model 2020 (EFSM20): geologic input data for the European Seismic Hazard Model 2020, Nat. Hazards Earth Syst. Sci. Discuss. [preprint], https://doi.org/10.5194/nhess-2023-118, in review, 2023

8. Lines 228 – 229: Why don't you use synthetic catalogs to reply to this question?

Thank you for this very interesting perspective. It is true that a more detailed study, possibly based on synthetic catalogs, would be a valuable continuation of this work. We have explored such an exercise and have submitted a separate paper to SRL dealing with synthetic catalogs.

9. Line 267: What is the interpretation for this?

A larger spatial scale smoothes the moment rate estimate. The geodetic signal has a larger spatial wavelength with respect to earthquake density distributions in space, in some cases the geodetic signal cannot capture some rapid spatial changes. Increasing the geographical where the comparison is done smooths this effect (see e.g. the discussion about 'small size areas' in Section 1.3.4 and 1.4).

10. Line 282: It is not the b–value but the a–value of the G–R law that expresses the level of seismic activity and the areal size. Please, comment on that and explain how you have adjusted the a–values and what the result has been.

Sure, the b-value is the slope of the Gutenberg-Richter distribution. We have modified the paragraph to avoid any confusion, as follows:

"In these areas, there are too few data to constrain the model,  the b-value is inferred from the larger macrozone, the a-value is inferred both from the larger macrozone and from the number of earthquakes in the area source zone (Danciu et al. (2021))."

Details on the building of the ESHM20 source model can be found in the EFEHR technical report Danciu et al. (2021), as well as in the 2024 NHESS article (https://doi.org/10.5194/egusphere-2023-3062).

11. Line 285: Could you be more specific about "unusual"? Besides, there are plenty of publications addressing the non–linearity of G–R relation in its entire range.

We have removed the term and we have modified the paragraph as follows: "In both area sources, the slope of the recurrence model in the upper magnitude range (mostly historical period) is lower than the slope in the moderate magnitude range (mostly instrumental period). This is not due to a lack of data. A double slope Gutenberg-Richter distribution has been used."

12. Line 308: These are defined by seismicity, faulting, and related physical properties. Have you used geodetic measurements alone to define seismogenic sources?

You refer to this sentence: "In those areas, the surface deformation measured by geodesy is therefore not a good proxy for the seismic activity and can not be used directly to constrain seismogenic source models."

Sorry for the confusion, we meant constraining earthquake rates, we have corrected the sentence : "In those areas, the  surface deformation measured by geodesy is therefore not a

suitable proxy for the seismic activity and can not be used directly to constrain earthquake rate models."

13. Lines 310–312: "… at least 30 events …" – small? Large? How have you selected these 30 events?

It is the total number of events used to infer earthquake recurrence parameters.

We have modified the sentence as follows: "Considering area sources with the best constrained recurrence models (at least 30 events used to estimate recurrence parameters, see Danciu et al. (2021)), the consistency between both moment rate estimates is strongly improved."

For more details, please refer to Danciu et al. (2021).

Line 399: is it only the number of earthquakes that matters or their magnitude (their moment respectively)?

Again, it is the total number of events used to model the recurrence (to infer a and b-values). We have modified the sentence as follows: "In regions of very low seismicity, with statistical fitting constraints relying on less than 10 events above the minimum magnitude of completeness, the distribution of the seismic moment …"

Line 402: Could you be more specific? How much "high" and how "episodic"?

We have deleted "episodic" and I have modified the sentence as follows: "There are exceptions, such as FRAS164 in the Western Pyrenees, a small zone with a high seismic activity with respect to the rest of the Pyrenees."

14. First paragraph of Conclusions section: It is rather a summary – please take it out if this section.
15. The last part of conclusions is rather "Discussion" than "Conclusions" – please provide explicitly the conclusions of the study.

We believe that the conclusion should provide a summary of the main findings, with perspectives and take home messages. This was an advice of reviewer #2, and the conclusion has now been completed following his /her suggestions. Thank you.

SPECIFIC COMMENTS

16. Section 1.2.2: It is better to put the calculation technique in an appendix.

We believe that it is easier to understand what is done when the method is clearly explained in the main text. Readers less interested in the method can always read faster those sections.

17. Line 178: In almost "inactive" areas have you considered crustal thickness from ambient noise tomography? Please, clarify. Caution: crustal thickness is not identical with seismogenic thickness.

Thank you for your comment. Based on the literature available on the use of strain rates to estimate seismic moment rates in comparable tectonic settings, we have identified a range for the effective seismic thickness, which is different from seismogenic thickness. To account for the large uncertainty associated with this parameter, we explore this range of values to obtain the geodetic moment rate distribution.

We have modified the paragraph:

'Whereas for the seismogenic thickness (H in Equations 6 to 8), we consider here the elastic thickness, i.e. the average thickness over which a region's principal faults store and release seismic energy (Ward, 1998). Only a fraction of the frictional slip takes place during earthquakes (Bird et al. 2002). Mazzotti et al. (2005) define the "effective seismic thickness" as the thickness of the crust where deformation is fully accommodated by seismicity. In an application in eastern North America, they show that this effective seismic thickness may represent only 40% of the seismogenic thickness based on maximum and minimum depths of earthquakes. The thickness considered in the literature to evaluate seismic moment release from strain rates usually varies between 10 and 15km. Pancha et al. (2006) used a fixed seismogenic thickness of 15km throughout the Basin and Range region in Western US. D'Agostino et al. (2014) applied a thickness of 10± 2.5km throughout the Apennines in Italy, whereas Stevens and Avouac (2021) considered 15km in the India-Asia collision zone. Carafa et al. (2017) estimated average coupled thicknesses between ~3 and ~8km for faults in Italy. As there is considerable uncertainty, we use three alternative values (5, 10, and 15 km) and propagate this uncertainty up to the geodetic moment rate estimates.'

18. Line 190: Could you, please, support why didn't you consider an area in Greece with high seismic activity? For the sake of comparison among areas with different strain rates.

We could have selected a source zone in Greece. These are example source zones, representative of what can be observed throughout Europe. We use these examples to explain the methodology. Results gathering all source zones are displayed later on in the manuscript.

19. Line 209: What do you mean by that term and how are you estimating it?

We copy-paste the sentence Line 209 : "Our aim is to compare the moment rate corresponding to the long-term ESHM20 earthquake rate forecast with the geodetic moment rate."

We are not sure about the term that you refer to in this comment. Can you be more specific? The earthquake rate forecast is another way of naming the source model for PSHA?

20. Lines 217–219: Lack of clarity, please rewrite this text.

The paragraph addresses the following : "The model thus relies on both recent observations (instrumental eq. catalogue) and past historical seismicity well as on a wider analysis of the seismogenic potential of the area. The earthquake rate forecast model also includes our current knowledge about active faults (fault traces, segmentation, extension at depth). Geodetic information has been used in some cases for estimating the deformation accumulating along these faults (Basili et al. (2023)). The strain model thus is not strictly independent from the source model, however GNSS velocities have not been directly used to build the ESHM20 source model. The strain rate model can be used to test the ESHM20 source model and evaluate how realistic the model is."

When observations are used to test a forecast model, one should always question if the observations have been used to build the model. So this is what we discuss rapidly in this paragraph.

21. Lines 221–224: this introductory part needs more elaboration.

The introduction of the paper now includes more references and is an introduction valid for this section too.

22. Line 254: what do you mean by that? Could you be more specific?

Lines 252-255 are the following:
"In some area sources such as GRAS257 in Greece, the mean geodetic moment rate results five times higher than the mean seismic moment rate and their distributions only partially overlap. In other source zones, such as FRAS176 in France or ITAS335 in Italy, the seismic and geodetic distributions are very consistent."
These are observations of what can be observed in Figure 9.

23. Line 255: Consistent on what? Could you be more specific?

Both histograms overlap quite well.

24. Line 261: Is the size of the source zone that matters or the deformation intensity?

The size of the area at which the comparison between geodetic and seismic moment rates is led, matters, as highlighted in different parts in the manuscript.

25. Line 262: Could you please explain briefly what are they and how are they defined?

The sentence in line 262 was "Macrozones are used as a spatial proxy in the building of the ESHM20 seismogenic sources".
We agree it is not clear, we have modified the sentence as follows: "Macrozones include several area source zones. They are used at different levels in the building of the ESHM20 seismogenic sources, e.g. to determine regional variations in the completeness of the catalog, or to define tectonic similarities and maximum magnitude throughout Europe (Danciu et al. 2024, Basili et al. 2024)."

26. Lines 349 – 351: Of course it does.

The sentence you mentioned in this comment is : "This suggests that the inclusion of active faults may strengthen the earthquake recurrence model in areas that are characterized by both a slow deformation rate and rare seismic events." We changed it to "This corroborates the idea that the inclusion of active faults may strengthen the earthquake recurrence model in areas characterized by both a slow deformation rate and rare seismic events."

27. Three first paragraphs of page 25: many repetitions – for the reader's sake please, reorganize the text.

We copy-paste the paragraphs you mentioned :

The figure 14a presents the ratio between the estimated $\dot{M}_{OS}$ and $\dot{M}_{OG}$ at the scale of source zones. The figure 14c provides a view of how these moments are distributed as a function of

the distance along the cross-section AB: the average geodetic ($\dot{M}_{OG}$) and seismic ($\dot{M}_{OS}$) moment rates are represented by plain orange bars and empty black bars, respectively. $\dot{M}_{OG}$ exceeds the mean $\dot{M}_{OS}$ in all source zones (5 to 10 times larger), except in the central source zone, which is the most seismically active and encompasses several faults. In this particular source zone (ITAS317), $\dot{M}_{OS}$ (computed as the weighted mean of all ESHM20 branches, shown by empty black bars) exceeds the geodetic moment estimated from strain rates.

We use the fault and smoothed seismicity model of ESHM20 source model logic tree to compare the seismic moments with the average geodetic moments from the strain rate solutions, evaluated on the same spatial grid. It should however be noted that the ESHM20's hybrid model forecasts seismic moments that are on average smaller than the models based on area source zones (figure 12, figure 14). The fault and smoothed seismicity model (purple bars) exhibits seismic moments that are systematically lower than the mean inferred from the full ESHM20 source model logic tree.

$\dot{M}_{OS}$ and $\dot{M}_{OG}$ are compared along a profile AB, averaged within spatial bins of 14 km (figure 14b). This analysis at a finer scale reveals that the seismic moment is concentrated on the fault traces (marked with small blue arrows). The geodetic moment rate exhibits a smoother behavior, and reaches its maximum ($4*10^{13}$N m yr$^{-1}$ km$^{-2}$) at the level of the eastern fault (similarly to $\dot{M}_{OS}$ ).

We changed these paragraphs into :

"Figure 14a presents the ratio between the estimated $\dot{M}_{OS}$ and $\dot{M}_{OG}$ at the scale of source zones. Figure 14c provides a view of how these moments are distributed as a function of the distance along the cross-section AB: the average geodetic ($\dot{M}_{OG}$) and seismic ($\dot{M}_{OS}$) moment rates are represented by plain orange bars and empty black bars, respectively. $\dot{M}_{OG}$ exceeds the mean $\dot{M}_{OS}$ in all source zones (5 to 10 times larger), except in the central source zone (ITAS317), which is the most seismically active and encompasses several faults. In this particular source zone, $\dot{M}_{OS}$ exceeds $\dot{M}_{OG}$ .

We use the fault and smoothed seismicity model of ESHM20 source model logic tree to compare the seismic moments with the average geodetic moments evaluated on the same spatial grid. It should however be noted that the fault and smoothed seismicity model (purple bars) exhibits seismic moments that are systematically lower than the mean inferred from the full ESHM20 source model logic tree (Figure 12, Figure 14).

$\dot{M}_{OS}$ and $\dot{M}_{OG}$ are compared along a profile AB, averaged within spatial bins of 14 km (Figure 14d). This analysis at a finer scale reveals that $\dot{M}_{OS}$ is concentrated on the fault traces, marked with small blue arrows. $\dot{M}_{OG}$ exhibits a smoother behavior, and reaches its maximum ($4*10^{13}$ N m yr$^{-1}$ km$^{-2}$) at the level of the eastern fault (similarly to $\dot{M}_{OS}$ )."

---

## Referee Report (RR1)

**Second Review of the manuscript egusphere-2024-787**

After reading this manuscript again, I am delighted to know its significant improvement. Most of the reviewers' comments have been considered. However, I still have some comments and questions, listed below. There are a few adjustments, which could improve the manuscripts.

In many cases, the reviewers' comments have been addressed by adding citations only and without going into details. I think the authors should explain some points better to make this manuscript a stand-alone paper which could be understood by the readers without checking the references. Some explanations, which are provided in the authors' response file, should be included in the manuscript, such as the response to Line 181, the response to Figure 4, and the response to Line 269 by the 2nd reviewer.

Some more points are the following.

1) The addition to the section conclusion is very little, it still seems to be more a summary than a section Conclusion. The strengths and limitations of the study could be better highlighted.
2) I would not completely agree that the comparison between this study and the geodetic model for Italy is outside the scope of this manuscript since it could validate and strengthen the analysis conducted here.
3) There are still editorial issues with the use of parentheticals for citing references, e.g (e.g. Zeng et al. (2018)) should be (e.g. Zeng et al., 2018) on line 56; (see Danciu et al. (2021)) should be (see Danciu et al., 2021) on line 101-102; Fault-Source Model 2020 EFSM20, Basili et al. (2023)) should be Fault-Source Model 2020 EFSM20, Basili et al., 2023) on Line 81; Mariniere et al. (2021)) should be Mariniere et al., 2021); etc.
4) I would suggest an English proofreading before the final version is ready to be published because the English language is quite poor in some parts. I report only some of them below.

Below there are a few (technical or editorial) comments on the manuscript.

Line 9: Replace "high activity zones" with "highly seismic activity zones".

Line 11: What does "local disparities underscore" mean? I would suggest rephrasing it.

Line 12: Replace "low-to-moderate activity zones" with "low-to-moderate seismic activity zones".

Line 20: replace "and its update the European" with "the updated European".

Line 21: Faults are actually included in the seismic source models in active countries, such as South Europe, Turkey, Japan, California, etc. So probably the sentence "In regions where active faults are rather well-characterized, they must be accounted for in the hazard estimations" should be corrected.

Line 26: replace "in the past, as the two" with "in the past, such as the two".

Line 33: Delete the second and third "on".

Line 41: Delete the second "if".

Lines 55-56: It is not enough to add only a citation to describe the Italian geodetic model. The authors should briefly explain what "the method of Carafa et al, (2017)" is. How does the

approach used to derive the Italian geodetic data differ from that used to derive the model of Piña-Valdés et al. (2022)? This is not beyond the scope of this manuscript since it gives the background and explains what already exists in the literature.

Line 56: Including one reference only contradicts "A number of studies…". Include more references. Also, did they use the same approaches?

Line 69: Delete "the" in "the most the compatibility".

Line 72: Replace "ESHM20 aims" with "ESHM20 aimed". Also, update the citation Danciu et al. (2021) with the peer-reviewed article Danciu et al. (2024) throughout the manuscript.

Line 73: Remove the space before the colon (in components :). This editorial typo appears often in the text, including in the caption of Figure 10 (seismic moment rate : 1 : ITAS308, 2 : ITAS331, 3 : ITAS339, 4 : BGAS043, 5 : FRAS164, 6: DEAS113, 7 : DEAS109, 8 : CHAS071 ).

Lines 74-75: "that" is repeated twice in the same sentence. I would suggest rephrasing it for better readability.

Line 87: Add "and" after the comma in "geologic features, seismicity pattern".

Line 89: Add "catalogue" before "completeness".

Line 91: Note that form 2 is not capital letter throughout the manuscript. The authors should check this.

Formula 1: Explain what Mmax,  N(m), a and b are. Also, does the minimum magnitude not appear in equation 1?

Line 94-95: It is still unclear what the corner magnitude is. Is the bending at large magnitudes? How does it relate to the minimum magnitude for the calculations of the recurrence parameters? It would be useful to include the formula perhaps.

Line 100: Leonard (2014) is an update of Leonard (2010) for stable continental regions only. Which one was used? Leonard (2010) for active shallow and subduction regions and Leonard (2014) for stable continental regions? The authors should be more precise. Also, replace "it" with "and".

Lines 100-101: I do not see more explanation added here to explain the smoothed seismicity model and the adaptive kernels. Adding only a citation is not enough to make this manuscript a stand-alone, independent paper. It would be useful to include the size of the spatial cells as indicated in the response file.

Line 108: Move "recurrence models" after Pareto.

Line 122: Replace "at the scale of Europe" with "at the European scale".

Line 127: Add a colon after "inverse problem" (without any space after "problem").

Line 132: Add a full stop before "however".

Line 134: Add ", i.e." after "categories".

Line 136: Delete finally.

Line 142: Leave a space between 142 and km. A space between a number and km should be checked throughout the manuscript.

Line 143: Change "the radius is increased, the" with "the radius increases, the".

Line 144: Add "is" after "this radius".

Line 147: Change "a number of decisions are required that may impact" to "a number of required decisions may impact".

Lines 149-150: Quadratic and Azimuthal should not have capital letters.

Formula 3: Explain what n is.

Line 167: Remove the comma before "uses".

Line 172: remove the comma before "propose".

Formula 7: What is AX?

Line 172: remove the comma before "uses" and remove the s in "uses". As indicated in my previous review, the use of the comma should be checked more carefully.

Line 179: Remove "focused" and add "of $C_g$" after two values.

Line 180: Replace "consider two values," with "consider two $C_g$ values,". It is unclear how the selected values of 2 and 2.6 were computed for dip = 25° and 65°. Since one of them is $C_g = 2$ as in Stevens and Avouac (2021), I assume that the dip angle should be also the same, i.e. 45°, and not 25°. Something here seems to be incorrect.

Lines 182-183: It is not enough to cite the work of Dziewonski and Anderson (1981) to justify the alternative values of the shear modulus. A brief explanation should be added here.

Lines 187-194: For the selection of the seismogenic thickness, using the case study of eastern North America does not seem correct for the highly seismic South Europe. Also, how to justify a thickness between 5 and 15 km for the seismicity with hypocentral depths of 20-25 km, which is present in Europe?

Line 204: Replace "the most" with mainly or mostly.

Line 215: Remove the comma after "variability".

Line 223: Replace "substantial" with "strong".

Lines 230-231: What is the impact of using different equations to calculate the geodetic moment since it is non-negligible? This sentence should be expanded.

Figure 5: In the caption replace "Full" with "Full distribution" as indicated in the x-axis. The labels in the y-axis of the top plots are still missing.

Line 245: Replace "realistic the model is" with "realistic ESHM20 is".

Line 257: Replace "stay" with "are" or "lie". Furthermore, this sentence is unclear because 1) the brackets are misplaced since it seems to be related to low-seismicity regions; 2) the sentence "seismic moment rates go down to much lower values" seems to be incomplete [than what??] and which regions are these seismic moment rates related to?

Lines 274-275: This sentence is unclear so the authors should rephrase it for better readability.

Line 299: The beginning of the first sentence should be rephrased because beginning with Let's is not suitable for a manuscript.

Line 305: " The source zones 305 CHAS071 (Switzerland), DEAS113 and DEAS109 (Germany) are not as active" [as WHAT??]. This sentence does not seem to be complete.

Line 311: Replace "is inferred both from the larger macrozone and from the number of earthquakes" with "is inferred from both the larger macrozone and the number of earthquakes". Also, the citation should be (Danciu et al., 2021) and should be replaced with the updated citation Danciu et al., 2024.

Lines 210-312: How was the final a value computed? Using the macrozones or the individual zones? This sentence could be improved to make clearer how the activity rate was estimated.

Lines 331-332: Include references for this sentence.

Lines 332-334: Include references for this sentence.

Line 370: Remove one bracket in "part))".

Lines 389-390: replace "As a consequence, and as seen at the scale of macrozones (Fig. 11), this discrepancy is reduced at a larger scale because of a spatial smoothing of the signal." with "This discrepancy is reduced at a larger scale because of a spatial smoothing of the signal as seen at the scale of macrozones (Fig. 11).".

Line 425: Replace "within" with "in".

Line 430-431: Why does FRAS164 behave differently from the surrounding zones?

Conclusions: Report examples of the regions when describing the results, for example, "in areas with small characteristic distances, such as XXX" in line 440; "In some of these areas, such as XXX-XX" in line 447; "whereas in others (e.g. XX-XXX)" in line 448; etc.

Lines 445-446: Include references for this sentence.

Lines 456-457: I don't find this sentence correct. In regions of slow deformation and low seismicity, it is often difficult to include the tectonic structures as fault sources in the seismic source model because the information on their geometry, the rupture behaviour, and the maximum magnitude they are capable of generating is incomplete or unknown. Furthermore, although the overall deformation rate in the region may be known, it is difficult to partition it among the active tectonic structures and thus estimate the activity rate of the individual faults.

References

Danciu, L., Giardini, D., Weatherill, G., Basili, R., Nandan, S., Rovida, A., Beauval, C., Bard, P.-Y., Pagani, M., Reyes, C. G., Sesetyan, K., Vilanova, S., Cotton, F., and Wiemer, S.: The 2020 European Seismic Hazard Model: overview and results, Nat. Hazards Earth Syst. Sci., 24, 3049–3073, https://doi.org/10.5194/nhess-24-3049-2024.

---

## Author Response (AR2)

Dear Editor,

Please find attached the second version of our revised manuscript. We have carefully considered all the comments.

The first review (RC2) is a detailed and constructive review. We did our best to account for most points raised and to incorporate most suggestions. When we do not incorporate the suggestion, we explain why. Sometimes the manuscript has been re-written.

The second review (RC3) is identical to its first version, the only request is to include 6 references. We looked into these articles and now provide a very short summary of them in our response. As these articles are not directly linked to our work, we had included only 1 reference in the 1st revised version of our manuscript. Because the reviewer insists, we now include 3 of them.

Responses to reviewers' comments can be found below. The questions are presented in black, and our answers are shown in green. Each modification to the manuscript text is indicated in blue, with the corresponding text of the revised manuscript provided.

In addition, all prior changes in the manuscript are highlighted in yellow, while new changes for the second review are highlighted in orange for easy reference.

Thank you for considering this revised version of our manuscript.

Sincerely,
Bénédicte Donniol Jouve on behalf of the co-authors

**RC2**
**Second Review of the manuscript egusphere-2024-787**

After reading this manuscript again, I am delighted to know its significant improvement. Most of the reviewers' comments have been considered. However, I still have some comments and questions, listed below. There are a few adjustments, which could improve the manuscripts.
In many cases, the reviewers' comments have been addressed by adding citations only and without going into details.
I think the authors should explain some points better to make this
manuscript a stand-alone paper which could be understood by the readers without checking the references. Some explanations, which are provided in the authors' response file, should be included in the manuscript, such as the response to Line 181, the response to Figure 4, and the response to Line 269 by the 2nd reviewer.

As a reminder, the question mentioned line 181 was :
"Line 181: Explain how 12 from "the 12 difference preprocessing parameters" comes from."
We modified the text as follows :
"Figure 4 displays the exploration tree set up to combine 12 different preprocessing parameters to filter the stations of the GNSS velocity fields (three selections of GPS stations times four outlier radii), with 12 different regularizations of the velocity fields inversion to determine strain rates (choice of the distance and spatial weighting scheme, choice of the weighting threshold) and with finally 36 different parameterizations to calculate the moment rate from the strain rates."

As a reminder, the comment for Figure 4 was :
"Figure 4: I would suggest including also the weights associated with each branch in Figure 4a and explaining how they were defined."
This is an exploration tree, not a logic tree as used in PSHA. Weights are not needed, we just combine all the alternative parameters. We fear that referring to weights might bring confusion. We made the following modifications in the caption of Figure 4a.
"Exploration of the tree results in 3x4x2x2x3x3x3x2x2=5184 alternative moment rate estimates."
and
"distribution of the geodetic moment rate estimates (histogram built from the 5184 values)"

As a reminder, the comment for line 269 was :
"Line 269: What are the reasons for the lack of good fit in Spain when the macrozones are used? Which are the specific criteria used to assess that the overall fit is good from Figure 11? I would say that the fit is relatively good only for the highly seismic regions, not for central and northern Europe looking at Figure 11."
We have re-written this part as follows:
"In regions of northern Europe, the overlap between geodetic and seismic moment distributions is low (below 35 per cent, in red).  A rather good fit is obtained for the

Euro-Mediterranean region (overlap above 35 per cent, in blue), except in Spain (Fig. 11). A more detailed analysis focusing on Spain would be necessary to understand why."

Some more points are the following.
1) The addition to the section conclusion is very little, it still seems to be more a summary than a section Conclusion. The strengths and limitations of the study could be better highlighted.

We agree that the conclusion needed to be re-written. Please find the new conclusion:

"Many studies have been published that study how GPS strain rate is correlated with changes in observed seismicity (e.g. Zeng et al. 2018). Far less studies have focused on the comparison between strain rates and long-term earthquake forecasts built for assessing probabilistic seismic hazard. These long-term earthquake forecasts rely strongly on past seismicity, whereas geodesy offers an independent view on the amount of deformation that might be released in the future. In the present study, we have compared the new European seismogenic source model (Danciu et al. 2024) with a strain rate model developed at the European scale (Piña-Valdès et al. 2022). The comparison is led in terms of moment rates.

For every area source zone of the ESHM20 source model, we have established a distribution for the geodetic moment rate, which accounts for uncertainties in the selection of GNSS stations, the calculation of the strain rates and the conversion into a moment rate. At the source zone scale, we compare the geodetic moment rate distribution with the seismic moment rate distribution, as inferred from ESHM20 source model logic tree. We show that the geodetic moment rate is rather well correlated with the seismic moment rate in the most seismically active regions of Europe (e.g. the Apennines, Greece, the Balkans, the Betics, southeastern France), whereas in the low seismicity regions the geodetic moment rate is much higher than the seismic moment rate (e.g. Parisian basin, northern and central Europe, Fennoscandia). Results show that both estimates are slightly more consistent when considering larger spatial regions. In moderate to high seismicity regions, the geodetic strain is in general representative of the current horizontal tectonic stresses. In the very low seismicity region of Fennoscandia, the geodetic signal might be dominated by glacial isostatic adjustment and the strain does not represent the long-term tectonic loading.

More work is needed to understand the consistencies or discrepancies obtained between strain rate based moments and moments relying on the long-term magnitude-frequency distributions built for PSHA. Some parameters such as the seismogenic thickness will need to be better evaluated to refine the estimation of the moment rate from strain rates. In regions of low seismicity where the geodetic moment rate appears disconnected from the seismic moment rate, for now this is not clear how geodetic data can contribute to establish long-term earthquake forecasts. However, in seismically active regions, our work demonstrates the strong correlation between long-term seismic moment rates and geodetic moment rates . In these regions, strain rates should be used to constrain earthquake forecasts for PSHA, either combined with earthquake catalog data, or as an alternative model independent from the earthquake catalog."

2) I would not completely agree that the comparison between this study and the geodetic model for Italy is outside the scope of this manuscript since it could validate and strengthen the analysis conducted here.

The present paper is aimed at comparing moments based on the Pina Valdes et al. (2022) strain rate models and those inferred from the long-term magnitude-frequency distributions used for calculating probabilistic seismic hazard at the scale of Europe (ESHM20, Danciu et al. 2024). We agree that it would be interesting to compare the strain model obtained in Pina-Valdès et al. (2022) at the European scale, with the strain rate model derived for Italy by N. D'Agostino and used in Meletti et al. (2021). If the Italian model was available, we would have included this comparison in the Pina-Valdès et al. (2022) article. However the Italian strain rate model has not been published yet. It is very briefly described in Visini et al. (2021), in section 3.3.4.1, as well as the gridded seismicity model derived from it, named MG1. Visini et al. (2021) only provide the gridded-seismicity rates (Fig. 8 in their paper, the figure is small and the details can't be seen, same as Fig. 3 in Meletti et al. 2021). No proper comparison can be led with our strain rate model. As for the second geodetically-based source model in Meletti et al. (2021), MG2, the underlying methodology is quite different and the comparison would not be straightforward. As for MG1, the Meletti et al. and Visini et al. papers only provide a brief description and the seismicity rates based on MG2.
We have augmented the introduction to provide details on the MG1 gridded seismicity source model based on strain rates (see below), as well as on the more complex MG2 model. We don't think we can do more at this stage.

3) There are still editorial issues with the use of parentheticals for citing references, e.g (e.g. Zeng et al. (2018)) should be (e.g. Zeng et al., 2018) on line 56; (see Danciu et al. (2021)) should be (see Danciu et al., 2021) on line 101-102; Fault-Source Model 2020 EFSM20, Basili et al. (2023)) should be Fault-Source Model 2020 EFSM20, Basili et al., 2023) on Line 81; Mariniere et al. (2021)) should be Mariniere et al., 2021); etc.
Done

4) I would suggest an English proofreading before the final version is ready to be published because the English language is quite poor in some parts. I report only some of them below.
We have reread the entire article and checked every sentence. Corrections performed are visible in the version of the manuscript with tracked changes.

Below there are a few (technical or editorial) comments on the manuscript.
Line 9: Replace "high activity zones" with "highly seismic activity zones".
Done

Line 11: What does "local disparities underscore" mean? I would suggest rephrasing it.
We have rewritten this part of the abstract.

Line 12: Replace "low-to-moderate activity zones" with "low-to-moderate seismic activity zones".
Done

Line 20: replace "and its update the European" with "the updated European".

Done

Line 21: Faults are actually included in the seismic source models in active countries, such as South Europe, Turkey, Japan, California, etc. So probably the sentence "In regions where active faults are rather well-characterized, they must be accounted for in the hazard estimations" should be corrected.

This sentence is within a paragraph that states that seismic hazard models include fault models in regions where faults are rather well-characterized (see below), so we don't fully understand this comment. Nonetheless, we have modified the sentence :
"they must be accounted for" => "they are accounted for".
in the following paragraph:
"Nowadays, source models in up-to-date probabilistic seismic hazard studies are based both on past seismicity and active tectonics datasets. For example, the source model logic tree in the European Seismic Hazard Model 2013 (Woessner et al. 2015) and the updated European Seismic Hazard Model 2020 (Danciu et al. 2024) include two main branches, an area source model and a fault model. In regions where active faults are rather well-characterized, they are accounted for in the hazard estimations (Stirling et al., 2014, Field et al. 2014, Beauval et al., 2018). Fault models are mostly based on geologic information, covering much larger time windows than the available earthquake catalogs."

Line 26: replace "in the past, as the two" with "in the past, such as the two".
Done

Line 33: Delete the second and third "on".
Done

Line 41: Delete the second "if".
Done

Lines 55-56: It is not enough to add only a citation to describe the Italian geodetic model. The authors should briefly explain what "the method of Carafa et al, (2017)" is. How does the approach used to derive the Italian geodetic data differ from that used to derive the model of Piña-Valdés et al. (2022)? This is not beyond the scope of this manuscript since it gives the background and explains what already exists in the literature.

We have significantly augmented the paragraph to include a description of the two geodetic source models used in Meletti et al. (2021) :
"However, to our knowledge, in Europe, the only seismic hazard model that integrates a source model based on strain rates is the new Italian hazard model (Meletti, 2021). The gridded-seismicity model MG1 (Visini et al. 2021) relies on a strain rate tensor field calculated using the VISR software (Shen et al. 2015), as in Piña-Valdés et al. (2022). The rate of seismic moment is converted into earthquake rates assuming that earthquakes are distributed according to a tapered Gutenberg-Richter. Two alternative seismogenic thicknesses are considered (7 and 13 km). The total seismic rate is scaled to the seismic moment release of the Italian catalog (Visini et al. 2021). Meletti et al. (2021) also includes a second more complex geodetically-based source model, MG2. In this case, the model relies on the NeoKinema code (Bird 2009) that delivers interseismic and long-term strain rates and velocities on a finite element grid (see Bird and Carafa 2016)."

We must underline that the strain rate model which relies on a methodology close to the Piña-Valdés et al. (2022) model, and that has been used to establish the gridded-seismicity model MG1, has not been published. It is only briefly described in Meletti et al. (2021) and Visini et al. (2021). The strain rates are not displayed, only the gridded-seismicity rates obtained. MG1 relies on strain rates that have been derived with a methodology comparable to Piña-Valdés et al. (2022). However, before conversion into seismic rates, the total seismic rate is scaled to the seismic moment release of the Italian catalog (Visini et al. 2021). A fraction of the geodetic moment is thus assumed aseismic, so that as a whole over Italy, the rates forecasted fit the observed rates. What we do in the present article is quite different. We simply compare the moments, based on the strain rate models, and based on the ESHM20 long-term MFDs. We do not use the strain rate model to establish a source model for PSHA.

Bird, P. (2009). Long-term fault slip rates, distributed deformation rates, and forecast of seismicity in the western United States from joint fitting of community geologic, geodetic, and stress direction data sets, J. Geophys. Res., 114, B11403, doi:10.1029/2009JB006317.

Bird, P. and M. M. C. Carafa (2016). Improving deformation models by discounting transient signals in geodetic data: 1. Concept and synthetic examples, J. Geophys. Res. Solid Earth, 121, doi:10.1002/2016JB013056.

Shen, Z.K., M. Wang, Y. Zeng and F. Wang (2015). Optimal Interpolation of Spatially Discretized Geodetic Data, Bull. Seismol. Soc. Am., 105, 2117-2127.
Visini et al. (2021), annals of geophysics,  doi:10.4401/ag-8608

Line 56: Including one reference only contradicts "A number of studies…". Include more references. Also, did they use the same approaches?
We have added the reference Kreemer and Youngs (2022), which is also on the relationship between strain rates and seismicity.
We have also added the references Riguzzi et al. (2012) and Farolfi et al. (2020) (review by RC3, the only comment of this reviewer is to add 6 references of articles on the relationship between observed seismicity and strain rates).

Kreemer, C., and Z. M. Young (2022). Crustal Strain Rates in the Western United States and Their Relationship with Earthquake Rates, Seismol. Res. Lett. 93, 2990–3008, doi: 10.1785/0220220153.

Line 69: Delete "the" in "the most the compatibility".
Done

Line 72: Replace "ESHM20 aims" with "ESHM20 aimed". Also, update the citation Danciu et al. (2021) with the peer-reviewed article Danciu et al. (2024) throughout the manuscript.
In some cases, we keep both the Danciu et al. (2001) EFEHR report reference, publicly available on the EFEHR website, and the article Danciu et al. (2024), as the EFEHR report contains information that is not in the peer-reviewed article. In the other cases, we modified the references.

Line 73: Remove the space before the colon (in components :). This editorial typo appears often in the text, including in the caption of Figure 10 (seismic moment rate : 1 : ITAS308, 2 : ITAS331, 3 : ITAS339, 4 : BGAS043, 5 : FRAS164, 6: DEAS113, 7 : DEAS109, 8 : CHAS071).
Done

Lines 74-75: "that" is repeated twice in the same sentence. I would suggest rephrasing it for better readability.
Done

Line 87: Add "and" after the comma in "geologic features, seismicity pattern".
Done

Line 89: Add "catalogue" before "completeness".
Done

Line 91: Note that form 2 is not capital letter throughout the manuscript. The authors should check this.
Done

Formula 1: Explain what Mmax, N(m), a and b are. Also, does the minimum magnitude not appear in equation 1?
We added the explanation in the text according to your advice :
"where N(m) represents the cumulative annual rate of events as a function of magnitude (m); a and b are the Gutenberg-Richter recurrence coefficients, respectively the productivity and the exponential coefficient ; and Mmax is the maximum magnitude."

Line 94-95: It is still unclear what the corner magnitude is. Is the bending at large magnitudes? How does it relate to the minimum magnitude for the calculations of the recurrence parameters? It would be useful to include the formula perhaps.
The corner magnitude is required by the well-known tapered Pareto distribution.
We have added the following sentence:
"With respect to the Anderson and Luco (1983) magnitude-frequency distribution, the sharp cutoff at a maximum magnitude in the truncated distribution is replaced by smooth tapering."

Line 100: Leonard (2014) is an update of Leonard (2010) for stable continental regions only. Which one was used? Leonard (2010) for active shallow and subduction regions and Leonard (2014) for stable continental regions? The authors should be more precise. Also, replace "it" with "and".
Leonard (2014), with title 'Self-Consistent Earthquake Fault-Scaling Relations: Update and Extension to Stable Continental Strike-Slip Faults', is an update of Leonard (2010) for stable continental regions as well as for active regions (interplate dip slip, interplate strike slip, SCR/intraplate dip slip, and SCR/intraplate strike slip).

Lines 100-101: I do not see more explanation added here to explain the smoothed seismicity model and the adaptive kernels. Adding only a citation is not enough to make this manuscript a stand-alone, independent paper. It would be useful to include the size of the spatial cells as indicated in the response file.

We have added the following description:
"The smoothed seismicity model is developed by optimizing the adaptive kernel bandwidth, the smoothing parameters and the declustering parameters. Training and validation sets are used to determine the optimal combination of parameters. Details are provided in the EFEHR report (see, Danciu et al. 2021)."

Line 108: Move "recurrence models" after Pareto.
Done

Line 122: Replace "at the scale of Europe" with "at the European scale".
Done

Line 127: Add a colon after "inverse problem" (without any space after "problem").
Done

Line 132: Add a full stop before "however".
Done

Line 134: Add ", i.e." after "categories".
Done

Line 136: Delete finally.
Done

Line 142: Leave a space between 142 and km. A space between a number and km should be checked throughout the manuscript.
Done

Line 143: Change "the radius is increased, the" with "the radius increases, the".
Done

Line 144: Add "is" after "this radius".
Done

Line 147: Change "a number of decisions are required that may impact" to "a number of required decisions may impact".
Done

Lines 149-150: Quadratic and Azimuthal should not have capital letters.
Done

Formula 3: Explain what n is.
Thank you for your careful review. This was indeed an error; *ncell* and *n* represent the same value in Equation 3. We have revised the equation accordingly.

Line 167: Remove the comma before "uses".
Done

Line 172: remove the comma before "propose".
Done

Formula 7: What is AX?
This was a misunderstanding: we had written MAX, not AX. We have adjusted the formula to enhance clarity.

Line 172: remove the comma before "uses" and remove the s in "uses". As indicated in my previous review, the use of the comma should be checked more carefully.
Done

Line 179: Remove "focused" and add "of Cg" after two values.
Done

Line 180: Replace "consider two values," with "consider two Cg values,". It is unclear how the selected values of 2 and 2.6 were computed for dip = 25 and 65. Since one of them is Cg =
2 as in Stevens and Avouac (2021), I assume that the dip angle should be also the same, i.e. 45, and not 25. Something here seems to be incorrect.
Done

Lines 182-183: It is not enough to cite the work of Dziewonski and Anderson (1981) to justify the alternative values of the shear modulus. A brief explanation should be added here.

We have included Burov (2011) to support the 30 GPa reference alongside the 33GPa proposed by Dziewonski and Anderson (1981). We have modified the text as follows :
"The uncertainty on the shear modulus is also taken into account, including two alternative values proposed for continental crust : $3.3 * 10^{10}\, N.m.yr^{-1}.km^{-2}$ and $3.0 * 10^{10}\, N.m.yr^{-1}.km^{-2}$ (e.g. Dziewonski and Anderson, 1981 ; Burov 2011) and widely used in the literature (e.g. Stevens et Avouac 2021; Working Group on California Earthquake Probabilities, 1995; Mazzotti and Adams, 2005)"

The shear modulus μ is commonly considered as the bulk shear modulus for the crust. For continental regions, two standard reference values—33 GPa and 30 GPa—are commonly used (Dziewonski and Anderson, 1981; Burov, 2011). These values are those used in the literature (e.g., 33 GPa Stevens and Avouac, 2021; and 30 GPa in Ward, 1998a; Working Group on California Earthquake Probabilities, 1995).

Burov, Evgene B. (Aug. 2011). "Rheology and strength of the lithosphere". In: *Marine and Petroleum Geology* 28.8, pp. 1402–1443

Lines 187-194: For the selection of the seismogenic thickness, using the case study of eastern North America does not seem correct for the highly seismic South Europe. Also, how to justify a thickness between 5 and 15 km for the seismicity with hypocentral depths of 20-25 km, which is present in Europe?

We mention the work of Mazzotti et al. in Western Canada for introducing the concept of effective seismic thickness, which is smaller than the seismogenic depth. The effective seismic thickness is not the total thickness over which earthquakes occur.

We then provide examples of the thicknesses used in different published works similar to our work, in Western US (15km), in Italy (10 km, 3 and 8 km), in the India-Asia collision zone (15km). All the studies mentioned are using strain rates to infer the moment that is released in earthquakes.

The thicknesses we use are within the range of the thicknesses used by previous studies in similar tectonic contexts.

To complement the paragraph, we have added the following sentence:

"Using strain rates to forecast earthquakes in the Italian seismic hazard model, Visini et al. (2021) assume elastic thickness equal to 7 and 13 km throughout Italy). As there is considerable uncertainty on this parameter, based on this literature review, we use three alternative values (5, 10, and 15 km) and propagate this uncertainty up to the geodetic moment rate estimates."

Line 204: Replace "the most" with mainly or mostly.
Done

Line 215: Remove the comma after "variability".
Done

Line 223: Replace "substantial" with "strong".
Done

Lines 230-231: What is the impact of using different equations to calculate the geodetic moment since it is non-negligible? This sentence should be expanded.
We have modified the sentence as follows:
"The results also highlight that the equation used to convert surface strain into scalar moment rate can have a significant impact (in blue in Fig. 5). The uncertainty on the choice of the equation contributes to the overall variability of the moment rate."

Figure 5: In the caption replace "Full" with "Full distribution" as indicated in the x-axis. The labels in the y-axis of the top plots are still missing.
Done

Line 245: Replace "realistic the model is" with "realistic ESHM20 is".
We are addressing only the source model of ESHM20, not the hazard results.
We correct the sentence: 'how realistic it is.'

Line 257: Replace "stay" with "are" or "lie". Furthermore, this sentence is unclear because 1) the brackets are misplaced since it seems to be related to low-seismicity regions; 2) the sentence "seismic moment rates go down to much lower values" seems to be incomplete [than what??] and which regions are these seismic moment rates related to?
"stay" has been replaced by "lie".

We have suppressed the sentence "Besides, we observe that in low-seismicity regions, geodetic moment rates lie between $\simeq 10^{11}$ and $10^{12}\ N.m.yr^{-1}.km^{-2}$ (in blue and green, in mainland Spain and France, northern Europe and Fennoscandia) whereas the seismic moment rates go down to much lower values."
Figure 8 is much more relevant for discussing what is observed in low seismicity regions.

Lines 274-275: This sentence is unclear so the authors should rephrase it for better readability.
We have re-written the sentence : "The distribution for the seismic moment is built by exploring the ESHM20 source model logic tree, taking into account the weights associated to each branch."

Line 299: The beginning of the first sentence should be rephrased because beginning with Let's is not suitable for a manuscript.
We changed the sentence into 'In eight area sources, the geodetic moment rate is, on average, significantly lower than the seismic moment rate.'

Line 305: " The source zones 305 CHAS071 (Switzerland), DEAS113 and DEAS109 (Germany) are not as active" [as WHAT??]. This sentence does not seem to be complete.
We added 'as FRAS164' for more clarity.

Line 311: Replace "is inferred both from the larger macrozone and from the number of earthquakes" with "is inferred from both the larger macrozone and the number of earthquakes". Also, the citation should be (Danciu et al., 2021) and should be replaced with the updated citation Danciu et al., 2024.
We have re-written the sentence, please see the next comment.

Lines 310-312: How was the final a value computed? Using the macrozones or the individual zones? This sentence could be improved to make clearer how the activity rate was estimated.
We have re-written the sentence:
"There are too few data to constrain the model,  the b-value is inferred from the larger macrozone, whereas the seismic activity is estimated by re-scaling the occurrence rates as a function of the number of complete earthquakes (the scaling factor is the ratio between the number of complete events in the area source and the number of complete events in the corresponding macrozone, Danciu et al. 2021)."
It is important here to keep the EFEHR report reference, because this level of detail is not in the peer-reviewed article.

Lines 331-332: Include references for this sentence.
The sentence you referred to is as follows:
"GIA generates a viscous asthenospheric flow and a large scale flexure of the overlying elastic lithosphere that results in rather large wavelength deformation (i.e. the strain is distributed over a large area)."
The citation of Mazzotti et al. (2011) and Piña-Valdes et al. (2022) has been added.

Mazzotti, S., A. Lambert, J. Henton, T. S. James, and N. Courtier (Dec. 28, 2011). "Absolute gravity calibration of GPS velocities and glacial isostatic adjustment in mid-continent North America: AG CALIBRATION OF GPS AND PGR". In: Geophysical Research Letters 38.24.

Lines 332-334: Include references for this sentence.
The sentence is as follows:
"It should also be noted that the postglacial rebound is a phenomenon that is not representative of the long term (a few Myrs) tectonics, but that it is a transient mechanism that started after the last glacial maximum, ≈ 20,000 years ago."
The citation of Steffen and Wu (2011), has been added.

Steffen, Holger and Patrick Wu (Oct. 1, 2011). "Glacial isostatic adjustment in Fennoscandia—A review of data and modeling". In: *Journal of Geodynamics* 52.3, pp. 169–204.

Line 370: Remove one bracket in "part))".
Done

Lines 389-390: replace "As a consequence, and as seen at the scale of macrozones (Fig. 11), this discrepancy is reduced at a larger scale because of a spatial smoothing of the signal." with "This discrepancy is reduced at a larger scale because of a spatial smoothing of the signal as seen at the scale of macrozones (Fig. 11).".
Done

Line 425: Replace "within" with "in".
Done

Line 430-431: Why does FRAS164 behave differently from the surrounding zones?
The case of the FRAS164 zone is discussed in Section 3.1.3.
We have modified the sentence as follows:
"There are exceptions, such as FRAS164 in the Western Pyrenees, a small zone with a high seismic activity with respect to the rest of the Pyrenees (as explained in Section 3.1.3)."

Conclusions: Report examples of the regions when describing the results, for example, "in areas with small characteristic distances, such as XXX" in line 440; "In some of these areas, such as XXX-XX" in line 447; "whereas in others (e.g. XX-XXX)" in line 448; etc.
We have re-written the conclusion and provide examples of regions.

Lines 445-446: Include references for this sentence.
The conclusion has been rewritten and we don't go into this detail in the new version.

Lines 456-457: I don't find this sentence correct. In regions of slow deformation and low seismicity, it is often difficult to include the tectonic structures as fault sources in the seismic source model because the information on their geometry, the rupture behaviour, and the maximum magnitude they are capable of generating is incomplete or unknown. Furthermore, although the overall deformation rate in the region may be known, it is difficult to partition it among the active tectonic structures and thus estimate the activity rate of the individual faults.

The sentence you mention is "the inclusion of active faults may therefore strengthen the earthquake recurrence model in areas that are characterized by both a slow deformation rate and rare seismic events."
We agree with you, we have suppressed this part of the conclusion.

References
Danciu, L., Giardini, D., Weatherill, G., Basili, R., Nandan, S., Rovida, A., Beauval, C., Bard, P.-Y., Pagani, M., Reyes, C. G., Sesetyan, K., Vilanova, S., Cotton, F., and Wiemer, S.: The 2020 European Seismic Hazard Model: overview and results, Nat. Hazards Earth Syst. Sci., 24, 3049–3073, https://doi.org/10.5194/nhess-24-3049-2024.

**RC3**
The reviewer asks for minor revision: "the article must include the most recent studies"
In his/her original review, the following studies were listed:
ART1 : Nakamura, M., Kinjo, A. Activated seismicity by strain rate change in the Yaeyama region, south Ryukyu. Earth Planets Space 70, 154 (2018).
ART2 : Pappachen, J. et al (2021). Crustal velocity and interseismic strain-rate in the Garhwal–Kumaun Himalaya. Scientific reports, 11(1), 1-13.
ART3 : Zeng, Y. et al (2018). Earthquake potential in California-Nevada implied by correlation of strain rate and seismicity. Geophysical Research Letters, 45.
In the paragraph 1.4 Focus in Italy, please produce a comparative and critical analysis with the following previous studies, focussing on the difference of the applied methods and the conclusion:
ART4 : Riguzzi, et al (2012). Geodetic strain rate and earthquake size: new clues for seismic hazard studies. Physics of the Earth and Planetary Interiors, 206.
ART5 : Farolfi, G., et al (2020). Spatial forecasting of seismicity provided from earth observation by space satellite technology. Scientific reports, 10(1), 1-7.
ART6 : Piombino, A. et al (2021). Assessing current seismic hazards in Irpinia forty years after the 1980 earthquake: Merging historical seismicity and satellite data about recent ground movements. Geosciences, 11(4), 168.

In the first version of our revised manuscript, we had added the reference Zeng et al. (2018). These papers are not directly related to our work, we are surprised that we are asked again to include all of them. Because we are in the second round of reviews, we have added two more articles, Riguzzi et al. (2012) and Farolfi et al. (2020), in the introduction. Please find below our explanations.

There are a large number of studies published studying the relation between strain rates and seismicity. We would like to underline that our study is different. We never compare the strain rate based moments with observed seismicity. We compare the strain rate based moments with seismic moments inferred from a probabilistic seismic hazard model (ESHM20). The seismic moments are obtained by integrating long-term magnitude-frequency distributions. These distributions are calibrated on observed seismic

rates (instrumental and historical earthquakes) and often extrapolated up to a maximum magnitude.

We have included the references that are directly related to our work, such as :

- Meletti et al. (2021) and Visini et al. (2021) are using a strain rate model for establishing a source model for probabilistic seismic hazard, we describe and cite this work.
- Carafa et al. (2017) developed a method to include strain rate in the building of a fault model for probabilistic seismic hazard assessment, the article is mentioned and cited.
- Jenny et al. 2004
- Stevens and Avouac 2021
- Mazzotti and Adams 2005

The 6 articles that the reviewer asks us to include are not comparing strain rates with long-term magnitude frequency distributions:

Nakamura and Kinjo (2018) evaluate the long-term strain rate by using GNSS data and compared it with seismicity activation during a 10 yrs period in the Ryukyu trench. They conclude that the long-term seismicity near Iriomote Island is strongly affected by changes in the crustal strain rate.  This paper could have been cited in the Pina-Valdes et al. 2022 article, but we don't know what would be the reason for absolutely including it in the present study ?

Papachen et al. (Scientific Reports 2021) analyze the interseismic strain rate using GNSS data in the Himalaya, to identify zones where large earthquakes could occur. They identify high compressional zones, extensional deformation zones, equal strain rate zones and low strain rate zones. They analyze crustal accommodation processes through the strain rate patterns. This study is only remotely close to our work.

Zeng, Petersen, Shen (2018). They correlate GPS strain rates with seismicity in California. They show that earthquakes of M > 6.5 are collocated with regions of highest strain rates, whereas smaller magnitude earthquakes of M ≥ 4 show clear spatiotemporal changes. They show that seismicity is closely related to the strain rate, and that, as the deformation field evolved out of the shadow in the late 1980s, strain has refocused on the major fault systems entering a period of increased risk for large earthquakes in California.

Riguzzi et al. (PEPI 2012) perform an analysis of the background strain rate in Italy and comparison with seismicity over a 5 years period. They conclude that the strain rate map may be a powerful tool for identifying the areas prone to the next earthquake.

Farolfi et al. (Scientific Report 2020) aims at understanding what controls the distribution of the seismicity. They use a strain rate field determined from GNSS data and satellite radar interferometry over a 20 years period. They study the correlation with M ≥ 2.5 earthquakes that occurred in the same period. They found that earthquake occurrence probabilities are linearly related to  strain rates. They conclude that strain rates can be used to forecast seismicity.

Piombino, Bernardi and Farolfi (Geociences 2021), entitled "Assessing Current Seismic Hazards in Irpinia Forty Years after the 1980 Earthquake: Merging Historical Seismicity and Satellite Data about Recent Ground Movements" use a strain rate map in Italy and show that there is a link between the strain rate and the shallow earthquakes, with their epicenters being placed only in high strain rate areas. The article analyzes the strain rate map with respect to the occurrence (or absence) of historical earthquakes.

---

## Author Response (AR3)

Dear Editor, Associate Editor, and Reviewers,

We would like to express our gratitude for the valuable and constructive comments provided. We have addressed most of the critical points raised.

Responses to reviewers comments can be found below. The questions are presented in black, and our answers are shown in blue.

In addition, all changes made in response to the first review are highlighted in yellow, while those related to the second review are highlighted in orange for easier reference.

Thank you for considering our answer.

Sincerely,
Bénédicte Donniol Jouve on behalf of the co-authors
* * *
This manuscript lacks significant innovation in its data, methodology, and results, which are critical for a high-impact publication.
In terms of data, the study relies heavily on existing models, such as those by Piña-Valdés et al. (2022) and Danciu et al. (2024), without presenting any novel or unique datasets that could advance the field of seismic hazard assessment. Furthermore, the study does not provide additional insights into previously underexplored or debated aspects of seismic and geodetic moment rate comparisons.

We are quite surprised to read such a sentence.
Since 1999, a number of seismic hazard models have been published at the scale of Europe (GSHAP, SESAME, ESHM13). This is the first time that the source model (earthquake forecast) of a European seismic hazard model is tested against a fully independent observation, strain rates based on GNSS. The ESHM20 model (Danciu et al. 2024) is tested against strain rates derived with the method of Piña-Valdés et al. (2022). Piña-Valdés et al. (2022) only produce a best estimate model. Here, we determine a distribution of strain rate estimates by exploring uncertainties at all steps in the strain rate estimation and their conversion into geodetic moment rates. As far as we know, our study is the first one that makes a comparison between seismic and geodetic moment rates at the scale of Europe, and that systematically tests and discusses the similarities and discrepancies for all source zone areas of the ESHM20 model. We believe that our results provide important insights for the community of scientists working on seismic hazard and active deformation in Europe.

From a methodological perspective, the analysis employs conventional approaches to compare geodetic and seismic moment rates. However, it fails to incorporate more advanced or state-of-the-art techniques, such as cascading rupture models or innovative inversion frameworks, which could significantly enhance the robustness and applicability of the results.

The strain rate model is obtained applying a state-of-the-art inversion methodology. We don't fully understand why cascading rupture models are mentioned. There are no cascading rupture models in probabilistic seismic hazard assessment, unless earthquake simulators are used, which is currently completely impossible to implement at the scale of Europe.

Regarding the results, while the manuscript identifies general agreements and discrepancies between geodetic and seismic moment rates in different regions, the findings largely reaffirm existing knowledge rather than delivering novel or transformative insights.

There are plenty of publications that compare geodetic and seismic moment rates. There are very few publications that compare the moment rates of a long-term earthquake forecast, built for PSHA, with respect to the moment rates inferred from geodetic strain rates. A long-term earthquake forecast is a model, often including an extrapolated part (e.g. extrapolation of the Gutenberg-Richter model in the upper magnitude range). It is of high interest to test the model against independent observations. Again, it is the first time an earthquake forecast built at the scale of Europe is compared to geodetic moment rates estimate build at the continental scale. This is one of the very few studies where earthquake recurrence models built for area source zones are tested against geodetic moment rates.

Key discrepancies, such as those observed in low-to-moderate seismic activity zones, are not explored in sufficient depth to yield actionable conclusions or guide future improvements in hazard modeling.

In the case of very low seismicity area sources, we observe poor or no correlation between moment inferred from the long-term earthquake recurrence model and geodetic moment. We write that in these areas, for now we believe that GNSS measurements cannot be used to constrain earthquake forecasts. We perform a comparison at the scale of Europe, this is true that more work is required to understand the differences obtained at local scale.

Although the authors have addressed some comments from the previous review round and made revisions, the core issues regarding the manuscript's originality, scientific contribution, and added value remain unresolved.
Overall, the manuscript's findings offer limited utility for advancing the understanding or practical application of strain rate and seismic hazard models.
Based on the aforementioned considerations, I recommend rejecting the manuscript for publication.

We must here remind the comments made by 3 reviewers :
RC1: "This is a quite interesting paper which focuses on the comparison of geodetic and seismic moment rates across Europe. The approach is successful in spite of the large area examined and its high seismotectonic heterogeneity.
RC2: "Although geodetic observations are not still routinely used to assess the seismic hazard in a region due to the lack of a long-term series of geodetic measurements, models based on geodetic observations have been shown to provide forecasting skills where traditional methods to assess seismic rate models have not (e.g., Rhoades et al., 2017; Rollins and Avouac, 2019; Gerstenberger et al. 2020). In this context, this manuscript is a step forward in this direction. "
RC4 : "The subject is fascinating and worth to be published."